# Stress-primed secretory autophagy promotes extracellular BDNF maturation by enhancing MMP9 secretion

Silvia Martinelli [1,18 ✉], Elmira A. Anderzhanova[2,3,18], Thomas Bajaj[2], Svenja Wiechmann [4,5,6], Frederik Dethloff[1,7], Katja Weckmann[1], Daniel E. Heinz [2,3], Tim Ebert[2], Jakob Hartmann [8], Thomas M. Geiger[9], Michael Döngi[10], Kathrin Hafner[1], Max L. Pöhlmann[11], Lee Jollans [1], Alexandra Philipsen[12], Susanne V. Schmidt[13], Ulrike Schmidt[14,15,16], Giuseppina Maccarrone[1], Valentin Stein [10], Felix Hausch [9], Christoph W. Turck[1], Mathias V. Schmidt [11], Anne-Kathrin Gellner[10,12], Bernhard Kuster [4,5,6,17] & Nils C. Gassen [1,2 ✉]

The stress response is an essential mechanism for maintaining homeostasis, and its disruption is implicated in several psychiatric disorders. On the cellular level, stress activates, among other mechanisms, autophagy that regulates homeostasis through protein degradation and recycling. Secretory autophagy is a recently described pathway in which autophagosomes fuse with the plasma membrane rather than with lysosomes. Here, we demonstrate that glucocorticoid-mediated stress enhances secretory autophagy via the stress-responsive co-chaperone FK506-binding protein 51. We identify the matrix metalloproteinase 9 (MMP9) as one of the proteins secreted in response to stress. Using cellular assays and in vivo microdialysis, we further find that stress-enhanced MMP9 secretion increases the cleavage of pro-brain-derived neurotrophic factor (proBDNF) to its mature form (mBDNF). BDNF is essential for adult synaptic plasticity and its pathway is associated with major depression and posttraumatic stress disorder. These findings unravel a cellular stress adaptation mechanism that bears the potential of opening avenues for the understanding of the pathophysiology of stress-related disorders.

[1] Department of Translational Research in Psychiatry, Max Planck Institute of Psychiatry, Munich, Germany. [2] Research Group Neurohomeostasis, Department of Psychiatry and Psychotherapy, University of Bonn, Bonn, Germany. [3] Department of Stress Neurobiology and Neurogenetics, Max Planck Institute of Psychiatry, Munich, Germany. [4] Chair of Proteomics and Bioanalytics, Technical University of Munich, Emil-Erlenmeyer-Forum 5, Freising, Germany. [5] German Cancer Consortium (DKTK), Munich, Germany. [6] German Cancer Center (DKFZ), Heidelberg, Germany. [7] Metabolomics Core Facility, Max Planck Institute for Biology of Ageing, Cologne, Germany. [8] Department of Psychiatry, Harvard Medical School and McLean Hospital, Belmont, MA, USA. [9] Institute for Organic Chemistry and Biochemistry, Technische Universität Darmstadt, Darmstadt, Germany. [10] Institut für Physiologie II, University of Bonn, Bonn, Germany. [11] Research Group Neurobiology of Stress Resilience, Max Planck Institute of Psychiatry, Munich, Germany. [12] Department of Psychiatry and Psychotherapy, University of Bonn, Bonn, Germany. [13] Institute of Innate Immunity, University of Bonn, Bonn, Germany. [14] Research Group Molecular and Clinical Psychotraumatology, Department of Psychiatry and Psychotherapy, Rheinische Friedrich-Wilhelms-Universität Bonn, Bonn, Germany. [15] Research Group Traumatic Stress & Neurodegeneration & PTSD Treatment Unit, Department of Psychiatry and Psychotherapy, University Medical Center Göttingen (UMG), Göttingen, Germany. [16] Department of Psychiatry and Neuropsychology, Maastricht University Medical Centre, School for Mental Health and Neuroscience, Maastricht, The Netherlands. [17] Bavarian Center for Biomolecular Mass Spectrometry, Freising, Germany. [18]These authors contributed equally: Silvia Martinelli, Elmira A. Anderzhanova. ✉email: silviamartinelli@outlook.de; nils.gassen@ukbonn.de

Excessive or prolonged stress represents a threat to homeostasis. To adapt to stress, several strategies evolved ranging from genetic and epigenetic mechanisms to the activation of molecular pathways, finally resulting in the modification of physiological and social responses. Failure in stress adaptation can lead to exaggerated stress responses that, in turn, can promote the development of numerous psychiatric disorders including major depressive disorder (MDD), conditions of pathological anxiety, post-traumatic stress disorder (PTSD), bipolar disorder, and schizophrenia[1–3].

A fundamental molecular mechanism of stress adaptation is the maintenance of proteostasis, which is regulated by two major systems of protein turnover: (1) lysosomal degradation, such as autophagy, and, (2) ubiquitin-proteasomal degradation. By maintaining the balance between synthesis and degradation, these two processes contribute to the regulation of synaptic connectivity and plasticity in the central nervous system (CNS)[4,5]. Different types of lytic autophagy with different strategies for substrate selectivity preserve a healthy neural environment by preventing accumulation of protein aggregates, controlling the quality and quantity of cellular organelles, and complementing immune functions such as pathogen defense[6]. Impairment of proteostatic regulation in the brain contributes to the development of proteinopathies and excessive inflammatory responses, hallmarks of neurodegenerative and neuropsychiatric diseases[7–9]. Cellular stressors such as starvation, oxidative stress, and infection are known threats to proteostasis, counteracted by autophagy. We previously reported that, similarly to cellular stressors, glucocorticoid (GC)-mediated stress leads to the activation of macroautophagy that is regulated by the stress-sensitive co-chaperone FK506-binding protein 51 (FKBP51, coded by *FKBP5*)[10].

Recently, a non-lytic type of autophagy, termed secretory autophagy, has been described[11]. In this pathway, starvation induces loss of lysosomal integrity, which derails the cargo proteins within autophagic vesicles to the plasma membrane for subsequent secretion into the extracellular milieu, instead of their degradation via lysosome-dependent lytic autophagy. Previous work has linked secretory autophagy to the immune response and inflammation[12]. Indeed, validated cargo proteins include potent inflammatory mediators such as cytokines and cathepsins[13–15]. Whether the main function of secretory autophagy is to discard the secreted proteins or relocate them for the exertion of extracellular functions remains yet unclear[12–15].

In this study, we investigated whether GC-mediated stress (also referred to as "stress" from here on) affects secretory autophagy, analogously to macroautophagy. Furthermore, we explored its effect on synaptic plasticity and thus as possible link to the pathophysiology of stress-related disorders. Our results demonstrate that GC-mediated stress enhances secretory autophagy via the co-chaperone FKBP51, and that matrix metalloproteinase 9 (MMP9) is among the regulated proteins leading to an extracellular increase in cleavage of pro-brain-derived neurotrophic factor (proBDNF) to its mature form (mBDNF) both in vitro and in vivo. Furthermore, we show that this stress-induced increase in extracellular mBDNF has a promoting effect on synaptic plasticity. Hereby, we unveil a mechanism linking autophagy, as GC-mediated stress response, to neuroplasticity—key features intimately connected to stress adaptation and stress-related disorders.

## Results

### FKBP51 mediates vesicle fusion underlying secretory autophagy.
We previously identified the co-chaperone FKBP51 as an essential molecular link between GC-mediated stress and macroautophagy[10]. In order to gain more insight into the FKBP51-directed molecular environment, we sought to identify interaction partners of FKBP51 by defining its interactome. In HEK293 cells, we overexpressed FLAG-tagged FKBP51 followed by immunoprecipitation (IP) and mass spectrometry analysis of immunoprecipitates and identified 29 interactors (Supplementary Table S1). To determine possible links of these interactors to proteostasis-relevant pathways, we subjected all 29 of them to an automated literature-mining screen using the terms "autophagy", "proteostasis", and "ubiquitin proteasome system" with a custom-developed Python algorithm (Fig. 1a). Among the proteins related to autophagy, the vesicular R-SNARE protein SEC22B piqued our interest due to its central function in the secretory autophagy pathway[16]. This release pathway comprises a stepwise succession of signaling proteins defined by three regulatory stages: (1) the disruption of lysosomal membranes by a stressor and the recruitment by galectins of a receptor, such as TRIM16, and its cargo; (2) the transport of the receptor-cargo complex to the autophagosomal membrane via the R-SNARE SEC22B; (3) the internalization of the cargo protein into the autophagosome followed by its fusion with the plasma membrane that is mediated by the complex formation of R- and Q-SNARE proteins (RQ-SNARE complex). This ultimately leads to the secretion of cargo proteins into the extracellular milieu. Before investigating the role of FKBP51 in any of these steps, we strove for replication of the interactome results by performing co-IP experiments in neuroblastoma SH-SY5Y cells. Indeed, we successfully replicated the association of FKBP51 with SEC22B (Fig. 1b, c) and with two additional FKBP51 interactors, RACK1 and UBC12 (Supplementary Fig. S1a). Next, insight into FKBP51's impact on secretory autophagy regulation was gained by examining the association of FKBP51 with other major players of this pathway: a putative secretory autophagy cargo, cathepsin D (CTSD)[16], its receptor TRIM16, and the autophagosomal membrane-spiking protein, LC3B. Neither CTSD nor LC3B could be identified as FKBP51 binding partners in co-IP experiments (Fig. 1d), suggesting that FKBP51 does not contribute to the transport of cargo proteins nor the decoration of autophagic vesicles by LC3B. TRIM16 was found to weakly bind to FKBP51, represented by a faint band, which we confirmed with a reverse co-IP and a pull-down experiment (Supplementary Fig. S1b, c). These intriguing results prompted us to further investigate a putative role of FKBP51 in cargo-receptor dynamics. Thus, we generated an FKBP51-deficient (*FKBP5* KO) SH-SY5Y cell line (Supplementary Fig. S1d) that enabled us to study the interaction of TRIM16 with CTSD and with SEC22B in the absence of FKBP51. Co-IP analyses revealed that FKBP51 does not affect the interactions of TRIM16 with its binding partners (Fig. 1e, f).

Glucocorticoid receptor (GR) activation enhances FKBP51 expression, which mediates the effects of stress on different cellular pathways and functions. We investigated whether GR activation influences the interactions between TRIM16 and CTSD or SEC22B by stimulating the cells with 100 nM of the GR agonist dexamethasone or vehicle for 4 h. Co-IP analyses showed that dexamethasone did not affect the association between TRIM16 and SEC22B nor between TRIM16 and CTSD (Fig. 1g, h).

The final step in secretory autophagy that allows for cargo secretion is the SNARE complex formation between the R-SNARE SEC22B and the $Q_{abc}$ SNARE complex, formed by the synaptosomal-associated proteins SNAP23 and SNAP29 and the syntaxins 3 and 4 (STX3/4). This event leads to the fusion of the autophagosome with the plasma membrane, and the subsequent release of cargo proteins into the extracellular milieu. Since we identified SEC22B as a (direct or indirect) binding partner of FKBP51, we proceeded to examine the impact of FKBP51 on the RQ-SNARE fusion process. Via co-IP analyses, we detected associations of FKBP51 with SNAP23, SNAP29 and STX3, and, to a lesser extent, also to STX4 (Fig. 1i). In addition, we found that dexamethasone treatment enhanced RQ-SNARE complex

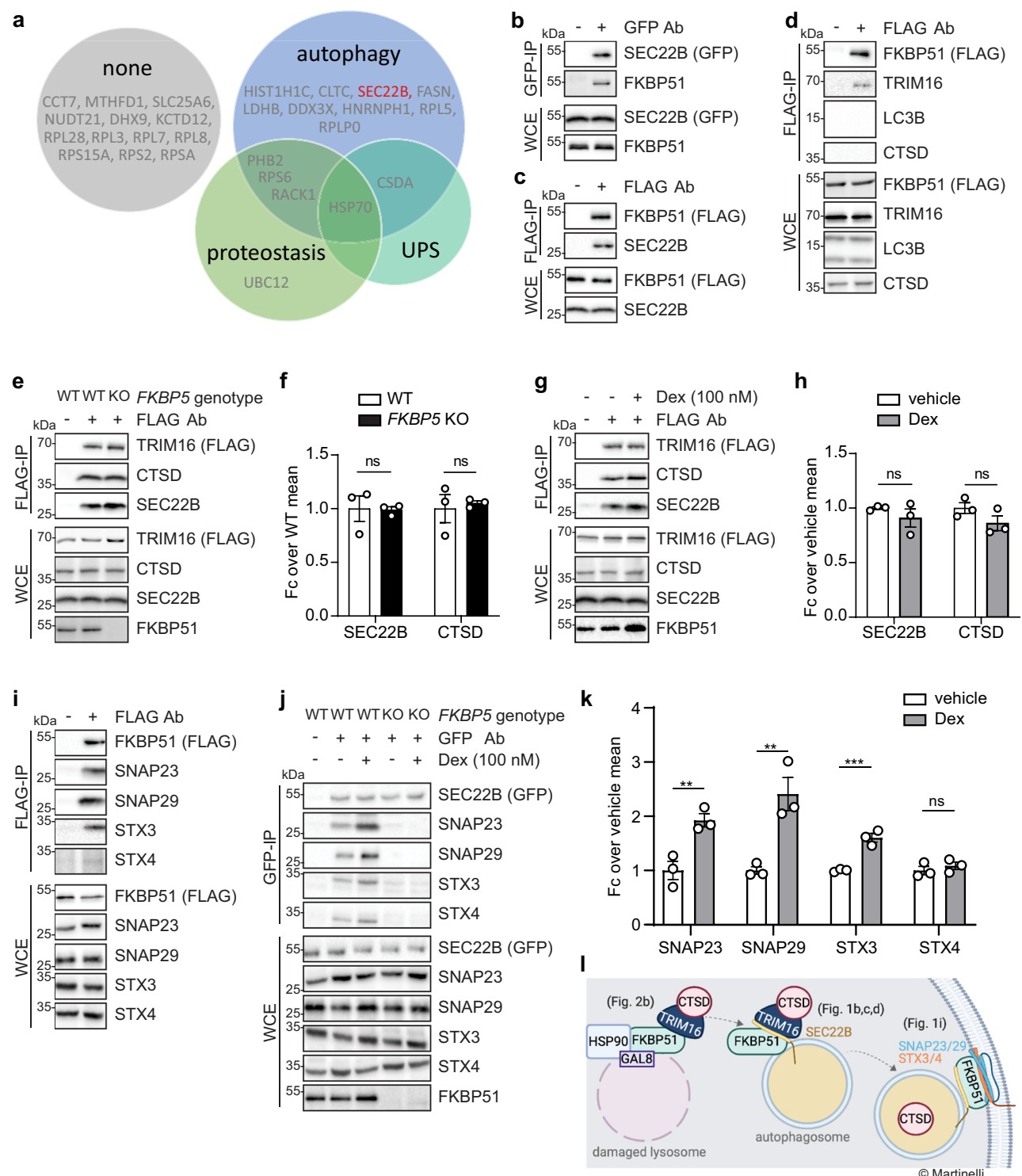

formation (Fig. 1j, k), presumably via FKBP51, whose expression was enhanced upon dexamethasone treatment (Supplementary Fig. S1e). Interestingly, co-IP analyses of SEC22B and Q$_{abc}$-SNARE proteins in *FKBP5* KO cells revealed that in the absence of FKBP51 the RQ-SNARE complex was almost abolished, both at baseline and after dexamethasone induction (Fig. 1j, k). From these data, we conclude a role for FKBP51 in several key steps of the secretory autophagy pathway (Fig. 1l).

**GCs induce lysosomal damage triggering secretory autophagy.** The reduction of lysosomal integrity is another hallmark of

secretory autophagy that plays a decisive role in targeting autophagosomes either to conventional lytic autophagy or to fusion with cytoplasmic membranes for non-canonical secretion[16]. In order to test if GC-mediated endocrine stress affects lysosomal integrity, we analyzed the effects of dexamethasone on the expression levels of galectin-3 (GAL3) and galectin-8 (GAL8), established markers of damaged endomembranes such as lysosomes as they act as danger receptors and recruit TRIM16[17–19]. Western blot analyses showed an overall increase in both GAL3 and GAL8 after dexamethasone treatment, suggesting enhanced lysosomal damage (Fig. 2a). Next, we determined whether FKBP51 associates with GAL3 or GAL8.

**Fig. 1 FKBP51 links stress to secretory autophagy. a** Results of automated literature mining of FKBP51 interactors in association to "autophagy", "proteostasis" and "ubiquitin-proteasome system" (UPS) or none of them. **b** Western blotting of FKBP51 and SEC22B in GFP-tagged SEC22B co-IP (GFP-IP) and whole-cell extract (WCE) as control; **c** Western blotting of FKBP51 and SEC22B in FLAG-tagged FKBP51 co-IP (FLAG-IP) and WCE as control. **d** Western blotting of FKBP51, TRIM16, LC3B, and CTSD in FLAG-tagged FKBP51 co-IP (FLAG-IP) and WCE as control. **e** Western blotting for TRIM16, CTSD, and SEC22B in FLAG-tagged TRIM16 co-IP (FLAG-IP) and WCE as control performed in WT and *FKBP5* KO SH-SY5Y cells. **f** Quantifications of **e** with $n = 3$ biologically independent samples. **g** Western blotting of TRIM16, CTSD, and SEC22B in FLAG-tagged TRIM16 co-IP (FLAG-IP) and WCE as control performed in cells treated with 100 nM dexamethasone or vehicle for 4 h. **h** Quantifications of **g** with $n = 3$ biologically independent samples. **i** Western blotting for FKBP51, SNAP23, SNAP29, STX3, and STX4 in FLAG-tagged FKBP51 co-IP (FLAG-IP) and WCE as control. **j** Western blotting of SEC22B, SNAP23, SNAP29, STX3, and STX4 in GFP-tagged SEC22B co-IP (GFP-IP) and WCE as control performed in WT and *FKBP5* KO cells treated with 100 nM dexamethasone or vehicle for 4 h. **k** Quantifications of **j** with $n = 3$ biologically independent samples. **b–k** All experiments were performed in SH-SY5Y cells. **l** Schematic model of the interactions of FKBP51 in the secretory autophagy pathway. Unpaired, one-tailed *t*-tests were performed for all quantifications; ns not significant, **$P < 0.01$, ***$P < 0.001$. Data shown as mean ± s.e.m. Ab antibody, Fc fold change.

Via co-IP, we identified GAL8 as a direct or indirect FKBP51 interactor, whereas GAL3 did not appear to bind to FKBP51. FKBP51 is a known interactor of the chaperone HSP90, which has been found to co-localize with GAL8[16]. Therefore, we hypothesized that the association between GAL8 and FKBP51 might be mediated by HSP90. Indeed, via co-IP analyses, we found HSP90 to associate with FKBP51, and observed a reduction in co-IPed GAL8 upon using a mutant form of FKBP51 lacking its ability to bind HSP90 (TPRmut, Fig. 2b). These results suggest that the association between GAL8 and FKBP51 is (at least) partially bridged by HSP90.

Then, to directly assess lysosomal damage, we used tfGal3 (tandem fluorescent-tagged Galectin3), an established reporter system employing an mRFP- and EGFP- tagged GAL3 in SH-SY5Y cells. While EGFP is degraded in acidic environment, mRFP is stable. This difference allows the monitoring of pH change in compromised lysosomes[20]: in case of lysosomal damage, the membrane becomes permeable (lysosomal membrane permeabilization), the acidic pH is lost and the labeled GAL3 proteins (tfGal3) are quickly recruited inside the lysosomes. Once the damaging stimulus ceases, the lysosomal membranes can be repaired and the acidic pH is restored. At this point, the EGFP of the tfGal3 constructs that have entered the lysosomes while they were damaged, are degraded. Therefore, a decrease of EGFP/mRFP ratio is indicative of lysosomal damage.

After transient transfection with tfGal3, various treatments were applied for 4 h and were subsequently washed off for different time periods, to allow lysosomal repair, as indicated in Fig. 2c and d. L-leucyl-L-leucine methyl ester (LLOMe) and bafilomycin A1 (Baf) were used for assay calibration[20]. LLOMe is a lysosome-damaging and inflammasome-activating substance and a well-characterized secretory autophagy activator[19] while Baf inhibits the vacuolar ATPase, thereby preventing lysosomal acidification[21]. Using fluorescence microscopy, EGFP⁺ and mRFP⁺ puncta were quantified and their ratio was determined. A reduction in EGFP⁺ puncta was detected upon LLOMe treatment, indicating a severe loss in lysosomal integrity. Co-application of Baf confirmed this mechanism, which resulted in no decrease of GFP puncta (Fig. 2c). Interestingly, dexamethasone caused a similar decrease in EGFP⁺ puncta in a dose- and time-dependent way (30 and 300 nM dexamethasone; 8 and 24 h after wash-off), indicating induced lysosomal damage (Fig. 2d). This effect was again suppressed by Baf (Fig. 2d). These results indicate that GR activation by GCs induces lysosomal damage.

Then, to determine the effect of GC-induced lysosomal damage on secretory autophagy, we analyzed CTSD release from cells by enzyme-linked immunosorbent assay (ELISA). A murine microglia cell line, SIM-A9, was chosen since microglia are the brain cells with the highest secretory capacity and for their relevance in the neuroimmune response. In this cell line, we first confirmed the GR responsiveness and consequential FKBP51 induction (Supplementary Fig. S2a) and, most importantly, the association

between FKBP51, SEC22B and the $Q_{abc}$-SNARE proteins (Supplementary Fig. S2b). We then investigated the effect of LLOMe- and dexamethasone-induced lysosomal damage on cargo secretion. In line with our previous results, the same treatments that caused an increase in lysosomal damage also led to enhanced secretion of CTSD (Fig. 2e, f), suggesting that GR activation triggers secretory autophagy. The underlying mechanism linked to lysosomal damage was further confirmed when we observed a decrease in secreted CTSD in case of co-treatment with Baf, compared to dexamethasone or LLOMe stimulation alone (Supplementary Fig. S2c).

**Identification of cargo proteins regulated by stress-induced secretory autophagy.** In order to capture the whole range of GC-mediated stress effects on secretory autophagy, and to identify the entire spectrum of cargo proteins released by this mechanism, we performed a secretome-wide analysis. We controlled for autophagy-dependent secretion of the analyzed proteins using a CRISPR-Cas9-generated autophagy-deficient microglial cell line (*Atg5* KO SIM-A9; Supplementary Fig. S3). We confirmed the functionality of this cellular model by quantification of secreted CTSD levels via ELISA. We observed enhanced secretion of CTSD in response to 100 nM dexamethasone in WT cells, while this increase was annulled in the absence of ATG5 (Fig. 3a), demonstrating that dexamethasone-induced secretory autophagy is tightly linked to ATG5-mediated signaling, and confirming the validity of this model.

For the secretome analysis, we used a proteomic method that combines metabolic labeling and click chemistry to selectively enrich and quantify released proteins[22]. Secretory autophagy was induced by treatment of WT and *Atg5* KO cells with dexamethasone. After LC-MS/MS measurement, database search and quality filtering, we were able to identify a total of 710 secreted proteins. A multiple *t*-test analysis (Fig. 3b; Supplementary Table S2) revealed that the secretion of a large number of proteins (75% of the total = 530 proteins) was enhanced in the WT sample compared to the *Atg5* KO upon dexamethasone treatment (FDR < 0.01). From this group, the vast majority (85% = 450 proteins) showed a change of at least 2-fold, while less than 4% of the detected proteins presented lower secretion levels in WT compared to *Atg5* KO. Interestingly, we did not only confirm an increase in secretion of CTSD, but also identified several other cathepsins that presented a strongly enhanced autophagy-dependent secretion (Fig. 3c), thereby uncovering secretory autophagy as a highly relevant release mechanism for members of the cathepsin protein family from microglial cells. Among the nine detected cathepsins, only cathepsin H (CTSH; also identified as pro-CTSH) displayed a lower secretion from the WT than from *Atg5* KO cells (Fig. 3c). For a better understanding of the autophagy-regulated secretome's functional profile, we performed a gene ontology (GO)

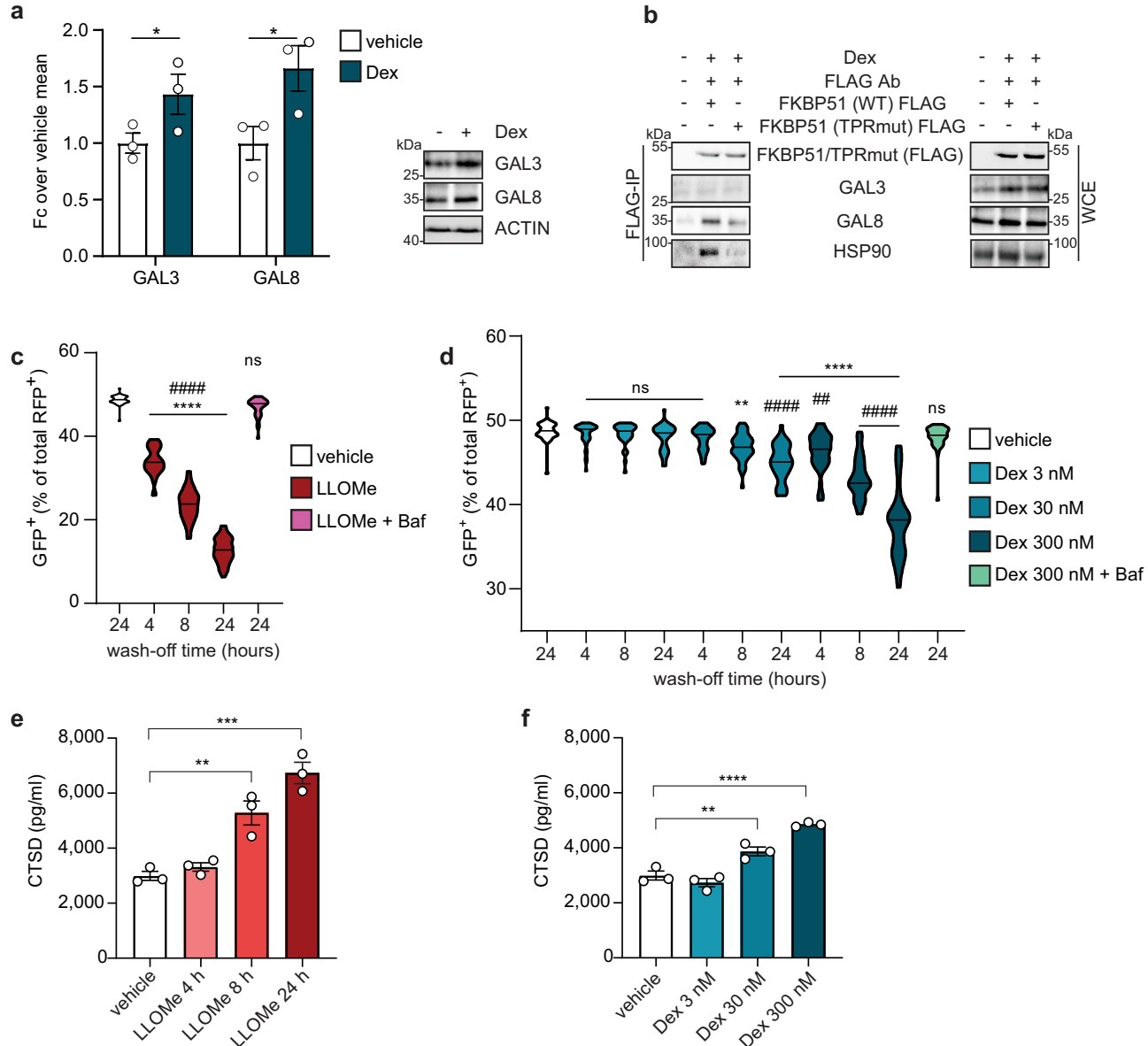

**Fig. 2 FKBP51 mediates regulatory mechanisms underlying secretory autophagy. a** Quantification of western blot analyses and representative blots of GAL3 and GAL8 normalized to actin from WT cells treated with 100 nM dexamethasone (Dex) or vehicle for 4 h. $n = 3$ biologically independent samples. Unpaired, one-tailed $t$-tests were performed; *$p < 0.05$. Data shown as mean ± s.e.m. **b** Western blotting of FKBP51, GAL3, GAL8, and HSP90 in FLAG-tagged WT or mutant (TPRmut) FKBP51 co-IP (FLAG-IP) and whole cell extract (WCE) as control performed in cells treated with 100 nM dexamethasone or vehicle. **c, d** Quantification of GFP$^+$ puncta expressed as a percentage of total RFP$^+$ puncta in cells ($n = 40$ cells per group) transfected with tfGal3 construct and treated with 1 mM LLOMe or 300 nM dexamethasone (Dex) for 3 h, followed by 4, 8, and 24 h wash-off, and with co-treatment of bafilomycin (Baf) for 3 h followed by 24 h wash-off. Kruskal–Wallis multiple comparison test was performed; **$P < 0.01$; ****$P < 0.0001$. Asterisk symbol indicates comparisons to vehicle; hash symbol indicates comparisons to treatment + Baf. **a**–**d** All experiments were performed in SH-SY5Y cells. **e, f** CTSD from supernatants measured via ELISA after SIM-A9 cells were treated with LLOMe for 4, 8, and 24 h or vehicle for 24 h, or with 3, 30, and 300 nM Dex or vehicle for 4 h. $n = 3$ per group. Tukey's multiple comparison test was performed; **$P < 0.01$; ***$P < 0.001$; ****$P < 0.0001$. Only significant comparisons are shown. Data shown as mean ± s.e.m. Fc fold change, Ab antibody.

analysis of the 450 proteins with increased secretion. The results displayed in Fig. 3d reveal a high functional heterogeneity of the analyzed set of proteins, covering most annotated pathways (full list in Supplementary Table S3). To pinpoint the role of GC-induced secretory autophagy in mechanisms underlying synaptic plasticity, we performed a synapse-specific GO analysis using the recently established SynGO analysis tool[23] (Fig. 3e; full results in Supplementary Table S4). 72 proteins were matched to SynGO annotated data sets, covering an interestingly high number of synapse-relevant proteins. Almost half of these proteins (33 of 72)

were found to be annotated in the cluster of synaptic organization, playing an essential role in structural reorganization and neurite growth, two key features of synaptic plasticity. To further verify these data and identify neuroplasticity-related proteins linked to stress-induced secretory autophagy, we performed an automated literature screening of the 450 analyzed proteins in association with neuroplasticity. Strikingly, 161 proteins have been found associated to neuroplasticity in at least one peer-reviewed publication (Fig. 3f; full results in Supplementary Table S5). For a better understanding of the identified

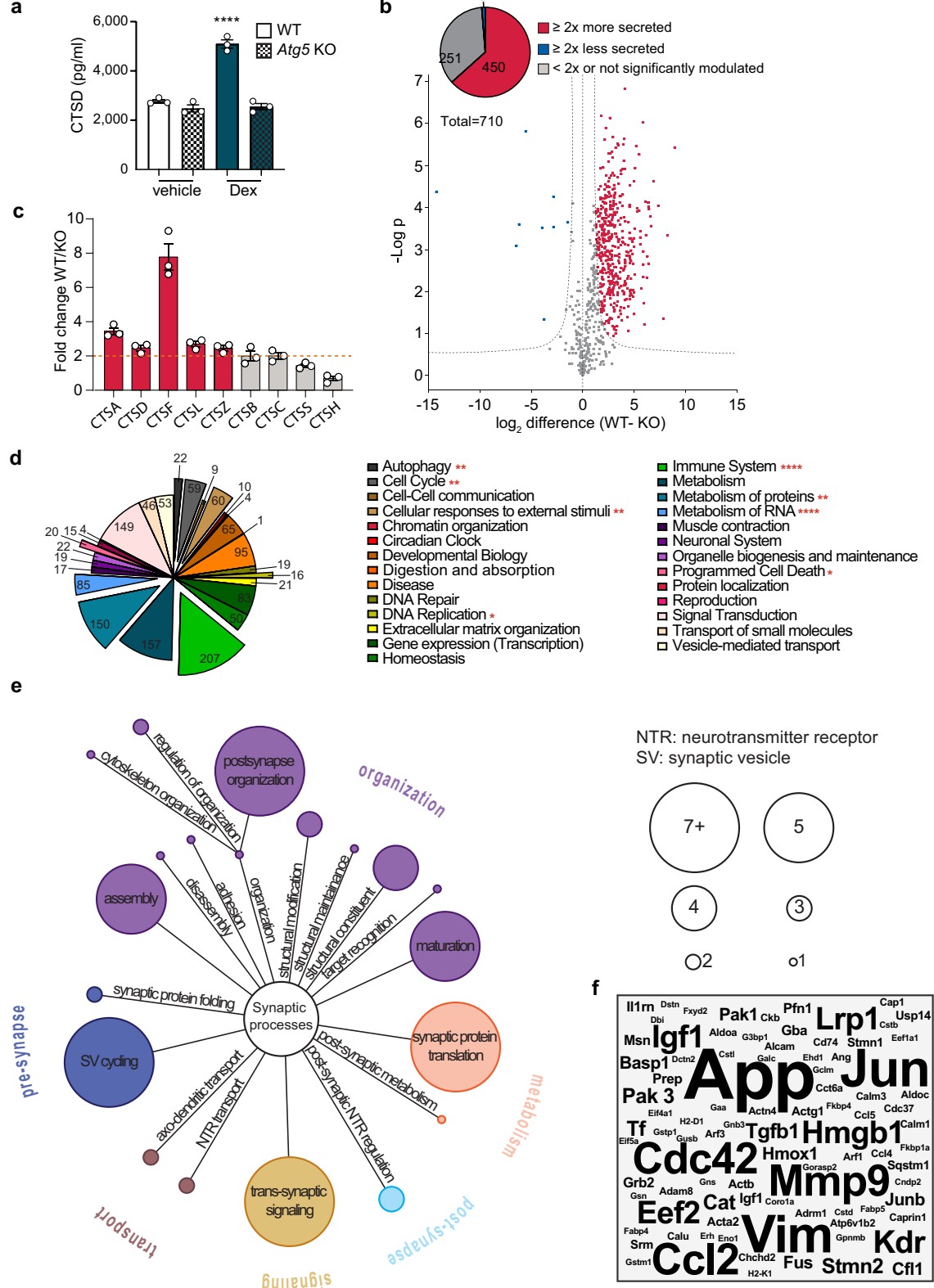

**Fig. 3 Detection of novel cargo proteins regulated by stress-induced secretory autophagy. a** CTSD from supernatants measured with ELISA after WT or *Atg5* KO SIM-A9 cells were treated with 300 nM dexamethasone (Dex) or vehicle for 4 h. Tukey's multiple comparison test was performed; ****$p <$ 0.0001. Significance refers to WT Dex compared to every other condition. Data shown as mean ± s.e.m. of $n = 3$ biologically independent samples. **b** Volcano plot representation of a multiple *t*-test analysis of the secretome in which each dot represents the replicates' mean (difference of WT-*Atg5* KO); FDR = 0.01, s0 = 1 of $n = 3$ biologically independent samples. **c** Secretion levels of detected cathepsins indicated as fold change of WT over *Atg5* KO. Data shown as mean ± s.e.m. of $n = 3$ biologically independent samples. **d** GO results performed with Reactome. **e** GO results performed with SynGO. **f** Word cloud representation of automated literature mining results in association with "neuroplasticity".

**Table 1 Neuroplasticity-related proteins regulated by stress-induced secretory autophagy.**

| Protein | Fold change | Functions | Related mental disorders |
|---|---|---|---|
| APP | | Neuronal maturation, synaptogenesis, synapse remodeling, neurite outgrowth, synaptic activity, LTP | AD |
| JUN | Detected in WT only | Neuronal differentiation and survival during development, post-traumatic repair, memory formation and addiction<br>Immediate early gene regulated by, rather than regulating, neuronal activity and plasticity | |
| VIM | | *Neuroplasticity and neuroregeneration* via *uptake of C3bot* | AD |
| CDC42 | | LTP, memory recall, neuronal maturation, synapse formation | |
| MMP9 | | Synaptic remodeling, *extracellular cleavage of proteins regulating synaptic plasticity*, late phase LTP, learning and memory | SCZ, FXS |
| CCL2 | | Spinal plasticity, short-term hippocampal synaptic plasticity, neuronal excitability and synaptic transmission via presynaptic mechanisms | AD, depression |
| EEF2 | | Synaptic plasticity, long term facilitation (LTF) and learning via local protein synthesis, key sensor for quality of neurotransmission; | |
| HMGB1 | | DAMP molecule, synaptic plasticity via inflammatory signaling | AD |
| IGF1 | | Hippocampal neurogenesis, social memory, LTP, age-related synaptic dysfunction | |
| KDR | Detected in WT only | Neurogenesis and cognition mediated by environmental stimuli, hippocampal excitatory synaptic function, plasticity and related emotional learning processes, hippocampal LTP and consolidation of contextual fear memory | |
| LRP1 | | Neuroregeneration, conductance of neuronal ion channels, NMDA-induced Ca2 influx into neurons, neurotransmission, NMDAR2B expression at the cell surface, synaptic integrity and function via GluA1 trafficking regulation, APP processing | AD |
| PAK1 | Detected in WT only | Inhibitory postsynaptic currents via GABA presynaptic releases, is regulated by cdc42 | AD, FXS, SCZ, HD |
| PAK3 | Detected in WT only | Mental retardation protein, synaptic transmission | Intellectual disability |
| STMN2 | | Axonal remodeling via late phase LTP microtubule turnover and presynaptic remodeling | |
| TF | | AMPA receptor trafficking efficiency and synaptic plasticity via its receptor (TFR) | |
| TGFB1 | | Synapsin distribution, synaptic depression, neurite outgrowth, LTP and memory | |
| CAT | | LTP, learning and memory via regulation of ROS | Depression, PTSD |
| HMOX1 | | Neuronal cytokinetics, survival and plasticity via brain sterol/oxysterol metabolism regulation | AD, PD |
| JUNB | | GR/MR induced hippocampal plasticity (putative)<br>Immediate early gene regulated by, rather than regulating, neuronal activity and plasticity | |

List of proteins and related functions found to be associated to neuroplasticity in at least five peer-reviewed publications. Bars indicate fold change in secretion of WT over Atg5 KO cells. Underlining indicates proteins also involved in immune response based on GO analysis; italic indicates extracellular function.
References used for this Table are listed in the Supplementary references.
*AD* Alzheimer's disease, *PD* Parkinson's disease, *SCZ* schizophrenia, *HD* Huntington disease, *PTSD* post-traumatic stress disorder, *FXS* fragile X syndrome.

neuroplasticity-related proteins, we manually analyzed and summarized the functions of the proteins mentioned in at least five publications reporting their relation to neuroplasticity (Table 1 and Supplementary references). Vimentin and MMP9 were of particular interest due to their reported extracellular function. Furthermore, over 70% of the top-ranked proteins (underlined in Table 1) were also linked to the immune system (a result of the previous GO analysis).

**FKBP51-mediated stress enhances BDNF maturation via extracellular MMP9.** MMP9 is an important regulator of BDNF signaling[24] and a proposed key factor of stress-related changes in the brain[25]. This knowledge motivated us to focus on this protein and to hypothesize that stress-induced release of MMP9 through secretory autophagy might promote the cleavage of proBDNF to mBDNF. First, to test this hypothesis we determined the levels of proBDNF and mBDNF as well as of MMP9 in the supernatants of WT and *Atg5* KO SIM-A9 cells exposed to dexamethasone. Both MMP9 and mBDNF were significantly more abundant in the supernatants of WT cells upon dexamethasone stimulation compared to vehicle, but this difference was absent in *Atg5* KO cells, as shown by ELISA (Fig. 4a, c). Conversely, proBDNF levels were slightly increased in response to dexamethasone, however in

an ATG5-independent manner (Fig. 4b). To exclude secondary effects of GCs (dexamethasone) on the secretion of BDNF and MMP9, we compared the effect of the more selective lysosomotropic agent LLOMe and, furthermore, tested different concentrations of dexamethasone (Supplementary Fig. S4a, b). These results confirmed that enhanced MMP9 secretion correlates with increased extracellular mBDNF but not proBDNF levels. In order to verify that this increase is determined by enhanced cleavage of proBDNF to mBDNF, rather than by increased mBDNF secretion, we repeated the experiments in the presence of the MMP9 inhibitor I (MMP9i). As expected, MMP9 secretion increased after dexamethasone treatment and was not affected by the presence of MMP9i (Fig. 4d), while proBDNF levels were unaffected both by the inhibition of MMP9 and by dexamethasone treatment (Fig. 4e). Conversely, while mBDNF levels increased after dexamethasone treatment, they were reduced upon coapplication of MMP9i (Fig. 4f). In summary, these results support our model; they propose that stress (dexamethasone) enhances MMP9 secretion leading to the conversion of proBDNF to mBDNF, thereby raising the mBDNF/proBDNF ratio by more than 2-fold compared to vehicle (Supplementary Fig. S4c).

Given that FKBP51 plays a central role in the regulation and mediation of stress on secretory autophagy, we tested the sole effect of increased FKBP51 on CTSD, MMP9, and BDNF

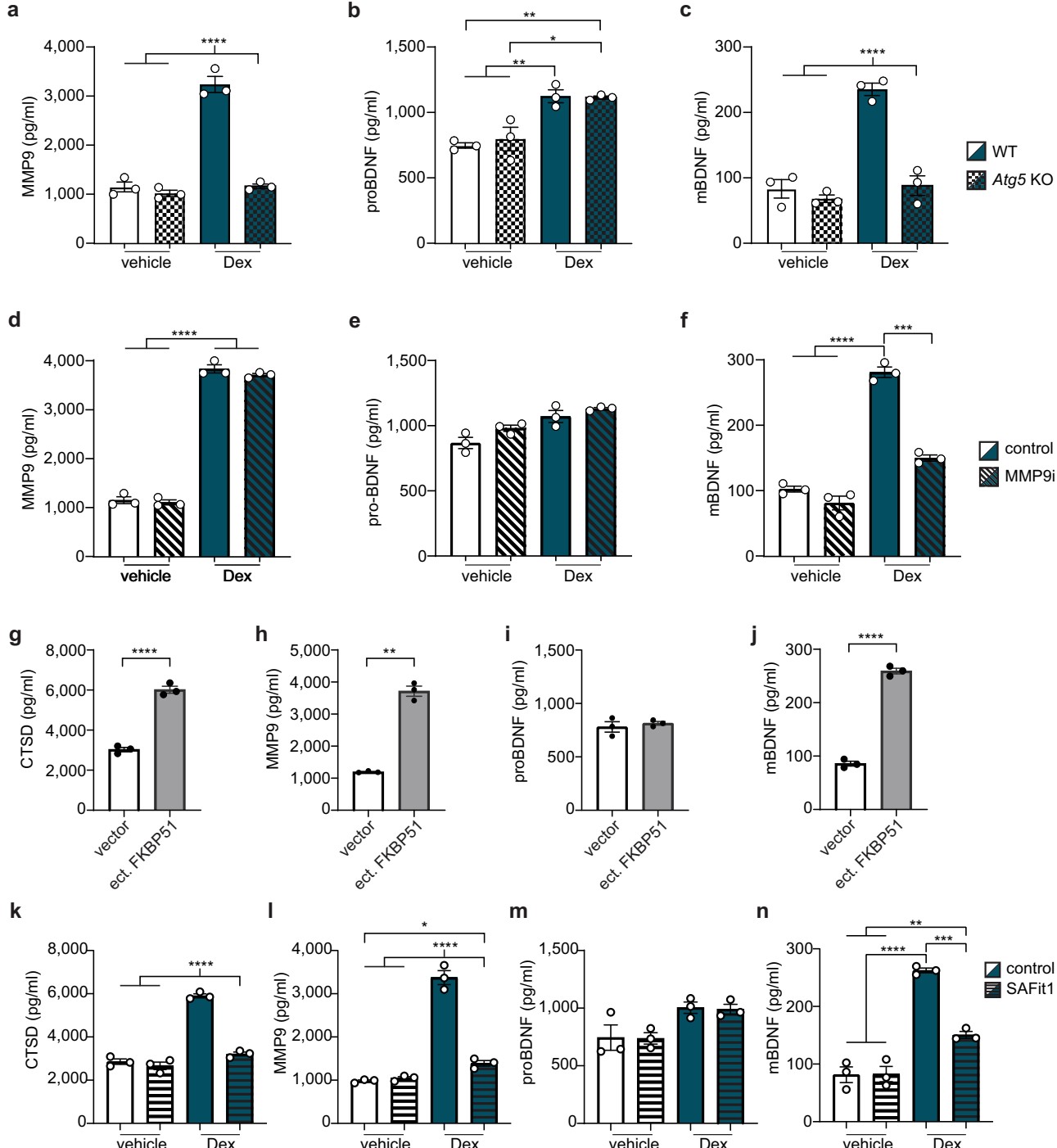

**Fig. 4 Stress enhances mBDNF production promoting proBDNF cleavage via MMP9. a** MMP9, **b** proBDNF, **c** mBDNF levels from supernatants measured via ELISA after WT or *Atg5* KO SIM-A9 cells were treated with 300 nM dexamethasone (Dex) or vehicle for 4 h. **d** MMP9, **e** proBDNF, **f** mBDNF levels from supernatants measured via ELISA after WT SIM-A9 cells were treated with 300 nM Dex, Dex + MMP9 inhibitor I (MMP9i), or vehicle for 4 h. **g** CTSD, **h** MMP9, **i** proBDNF, and **j** mBDNF levels from supernatants measured with ELISA after WT SIM-A9 cells were transfected with FKBP51 expressing plasmid (ect. FKBP51) or control vector. **k** CTSD, **l** MMP9, **m** proBDNF, **n** mBDNF levels from supernatants measured via ELISA after WT SIM-A9 cells were treated with vehicle, SAFit, 300 nM dexamethasone (Dex), and 300 nM Dex + SAFit for 4 h. For **d**, Mann–Whitney, one-tailed test; for **g**, **i**, **j** unpaired, one-tailed *t*-tests and for **h** unpaired, one-tailed Welch's test were performed; for all the others, Tukey's multiple comparison test was used; *$p < 0.05$; **$p < 0.01$; ***$p < 0.001$; ****$p < 0.0001$. Only significant comparisons are shown. Data shown as mean ± s.e.m. $n = 3$ biologically independent samples.

secretion: we overexpressed FKBP51 in SIM-A9 cells and analyzed the supernatants with ELISA. Interestingly, higher levels of FKBP51 (mimicking the effect of GR activation by stress) were sufficient to cause enhanced secretion of CTSD and MMP9 (Fig. 4g, h). Furthermore, levels of mBDNF but not proBDNF were increased in the presence of high levels of FKBP51, indicating an elevated MMP9-dependent cleavage rather than increased secretion of the neurotrophin itself (Fig. 4i, j). We additionally confirmed and corroborated the central role of FKBP51 via secretory autophagy, by generating two SIM-A9 cell lines lacking respectively FKBP51 (*Fkbp5* KO) and SEC22B (*Sec22b* KO; Supplementary Fig. S4d, e). Via ProteinSimple analyses, we validated that both FKBP51 and SEC22B are essential for stress-induced MMP9 secretion and BDNF maturation (Supplementary Fig. S4f, g). Having confirmed the key function of FKBP51 in stress-mediated secretory autophagy, we tested the effect of the FKBP51 antagonist SAFit1[26] on this mechanism. SIM-A9 cells were co-treated with SAFit1 or vehicle as well as dexamethasone or vehicle for 4 h. ELISA quantifications indicated that treatment with SAFit1 prevented the dexamethasone-induced CTSD and MMP9 secretion (Fig. 4k, l). Furthermore, SAFit1 significantly reduced the effect of dexamethasone on mBDNF but not on proBDNF levels (Fig. 4m, n). These results not only confirmed FKBP51 as a key player in secretory autophagy-mediated MMP9 secretion and subsequent BDNF maturation, but also revealed its potential as a drug target for the modulation of pathomechanisms underlying stress-related disorders.

**Stress enhances secretory autophagy and elevates extracellular mBDNF in vivo.** To understand the relevance of stress-induced secretory autophagy and its impact on brain physiology, we used microdialysis, a powerful neurochemistry approach for monitoring changes in extracellular content of endogenous or exogenous substances in animals in vivo (Fig. 5a). With this translation to a murine model, we aimed not only to achieve a better understanding of the secretion dynamics of selected proteins in a complex system, but also to validate the molecular mechanism we discovered in vitro, by application of a physiological stressor in living animals. We exposed mice to an acute electric foot-shock (FS; 1.5 mA, ×2) and quantitatively analyzed changes in selected proteins in the extracellular space of the medial prefrontal cortex (mPFC). To assess the effect of acute stress on protein secretion dynamics, microdialysate fractions were collected during baseline condition (at 120, 60, and 30 min prior to the FS), at the onset of the FS and after the FS (post-FS; at 30 and 150 min post FS) (Fig. 5b). To control for the dependency of the secreted proteins of interest on the FKBP51-regulated secretory autophagy pathway characterized in vitro, we compared secretion dynamics of WT and *Fkbp5* KO mice. The 30-min dialysate fractions were analyzed using capillary electrophoresis-based immunodetection (Wes, ProteinSimple). To verify this approach, we first analyzed the established secretory autophagy cargo CTSD and found increased CTSD levels after FS (Fig. 5c). This increase was diminished in *Fkbp5* KO animals. This confirmed our in vitro data showing that stress-induced secretion of CTSD is mediated by FKBP51. Furthermore, to corroborate our in vitro data on the regulation of BDNF maturation in vivo, we analyzed MMP9, proBDNF, and mBDNF in the microdialysates and found an increase in MMP9 and mBDNF levels after FS (Fig. 5d, f). Like the increase in CTSD, also this increase was diminished in *Fkbp5* KO animals. ProBDNF showed an increase only 150 min after FS, independently of FKBP51 (Fig. 5f). These results, first, confirmed that the modulation is specific for mBDNF, and therefore mediated by MMP9 (as demonstrated

also in vitro via selective MMP9 inhibition; Fig. 4d, e, f), and, second, revealed an interesting regulation of proBDNF by stress that is independent of secretory autophagy, in accordance with our in vitro data (Supplementary Figs. S4b, S5). To confirm that secretion of CTSD and MMP9 as well as the observed changes in extracellular mBDNF levels depend on the autophagic machinery, we measured their stress-dependent extracellular dynamics in mPFC microdialysates of C57BL/6 mice treated intraperitoneally with an established inhibitor of autophagy, the selective Unc-51 like autophagy activating kinase 1 (ULK1) inhibitor, MRT 68921[27] (ULK1i, 5 mg kg$^{-1}$). An increase of CTSD and MMP9 was determined in vehicle-injected mice after FS (Fig. 5g, h). This induction was significantly impaired in mice treated with ULK1i. The increase in MMP9 was paralleled by an autophagy-dependent increase of mBDNF after FS (Fig. 5j), while proBDNF showed a late-phase increase both in vehicle and ULK1i-treated mice (Fig. 5i). These data corroborate the in vitro findings showing that psychophysiological stress not only has an effect on autophagy-dependent secretion of CTSD and MMP9, but also leads to an increased extracellular mBDNF enrichment in murine mPFC.

**Stress-induced BDNF maturation affects synaptic plasticity ex vivo.** It is well established that BDNF is an important regulator of brain signaling and synaptic plasticity[28–30]. Given that stress-enhanced extracellular BDNF maturation via increased MMP9 secretion in the murine brain, we next investigated whether this (secretory autophagy-mediated) mechanism might consequently affect synaptic plasticity. Therefore, we treated organotypic hippocampal slice cultures (OHSC) of Thy1 (or Cluster of Differentiation 90, CD90)-GFP-expressing mice with dexamethasone or vehicle for 4–6 h. For the last 30 min, either MMP9i or vehicle was added to the medium. Subsequently, neurons of the CA1 were imaged twice at an interval of 30 min (t0 and t30) and at the end media were collected (Fig. 6a). First, we analyzed the culture media for the secretion of CTSD, MMP9, proBDNF, and mBDNF. Confirming the cell culture and in vivo microdialysis results, analyses of the culture media showed that increasing concentrations of dexamethasone led to increased secretion of both CTSD and MMP9 (Supplementary Fig. S6a, b), while MMP9i treatment had no effect on the dexamethasone-induced enhanced secretion (Fig. 6b, c). In addition, neither dexamethasone stimulation nor MMP9i treatment induced any changes in secreted proBDNF from OHSC (Fig. 6d, Supplementary Fig. S6c). In contrast, mBDNF levels were progressively enhanced with increasing concentrations of dexamethasone, but returned back to the baseline levels with the addition of MMP9i (Fig. 6e, Supplementary Fig. S6e). Together, these results confirmed a stress-induced increase of extracellular mBDNF (and increased mBDNF/proBDNF ratio; Supplementary Fig. S6e) in OHSC. Therefore, we next assessed whether this mechanism might be able to alter the spine density of dendrites from CA1 neurons of the OHSC using two-photon microscopy (Fig. 6f). The largest change in spine density over the 30 min imaging interval was observed in the vehicle-treated slices, while increasing concentrations of dexamethasone had a more stabilizing effect (e.g. increased proportion of unchanged spines, Fig. 6g). This effect was exacerbated by the addition of MMP9i (Fig. 6g) and paralleled by a progressive reduction in the proportion of reduced spine density within the "changed" subpopulation (Supplementary Fig. S6f). Interestingly, however, we could observe an overall increase in spine density upon dexamethasone treatment, while the addition of MMP9i resulted in a return to baseline level (Fig. 6h). These results suggest that secretory autophagy-mediated increase of extracellular mBDNF correlates with an increase in spine formation that can be reversed by inhibiting MMP9.

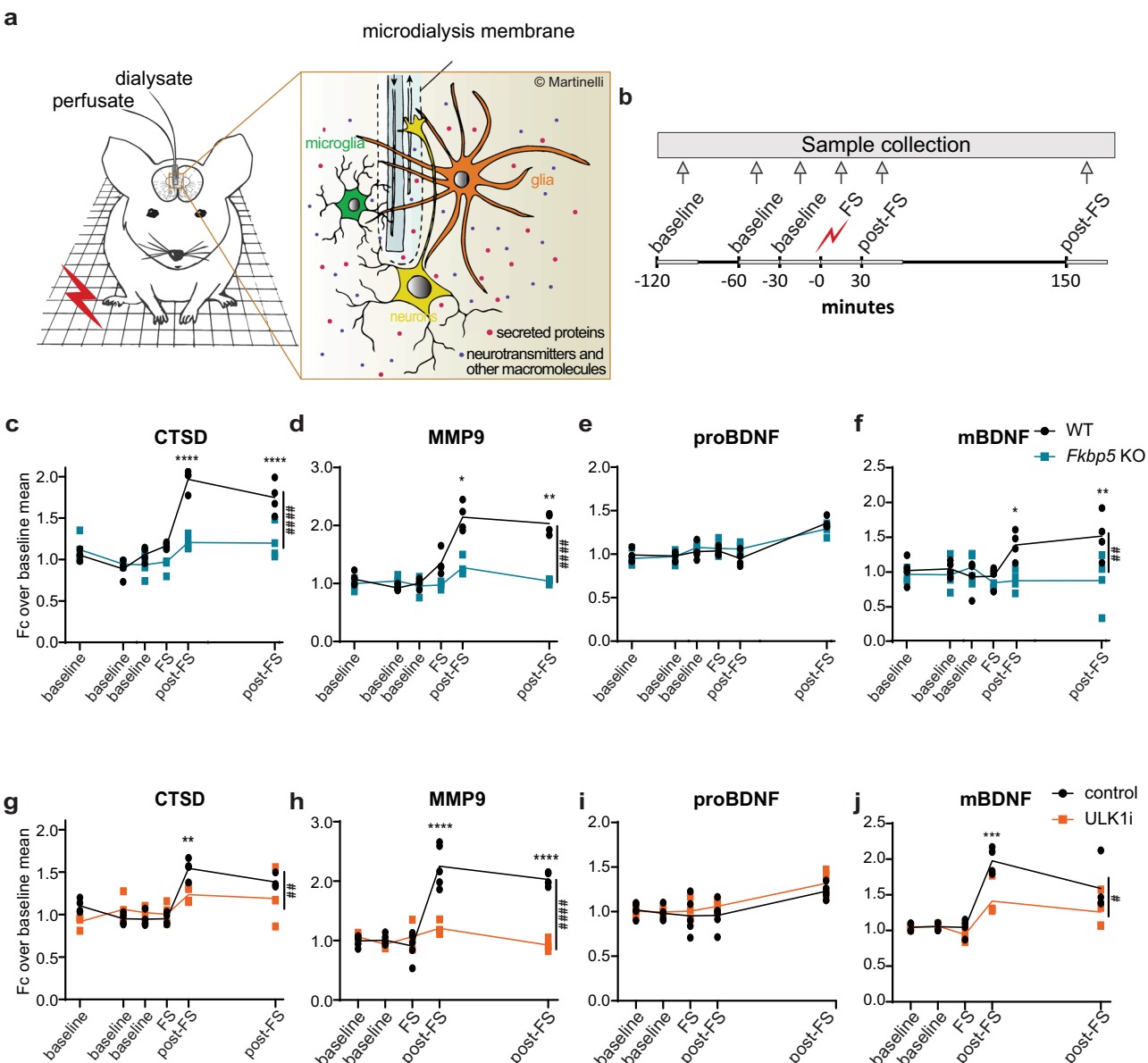

**Fig. 5 Stress enhances secretory autophagy and increases the mBDNF/proBDNF ratio in vivo. a** Schematic overview of in vivo microdialysis. **b** Experimental design and timeline; each sample was collected over 30 min indicated by the light gray lines. Quantifications of **c** CTSD, **d** MMP9, **e** proBDNF, and **f** mBDNF from in vivo mPFC microdialyses of WT and *Fkbp5* KO mice. Quantifications of **g** CTSD, **h** MMP9, **i** proBDNF, and **j** mBDNF from in vivo mPFC microdialyses of WT mice injected intraperetoneally with ULK1 inhibitor (ULK1i) or saline. *n* = 4 mice per group. Two-way ANOVA with Šídák's multiple *t*-test was performed; *$P < 0.05$; **$P < 0.01$; ***$P < 0.001$; ****$P < 0.0001$. Hash symbol refers to time × genotype/treatment interaction. Data shown as mean ± s.e.m. FS foot shock, Fc fold change.

## Discussion

In this study we describe a novel mechanism in which GC signaling and acute FS stress enhance secretory autophagy in an FKBP51-dependent manner, leading to increased extracellular BDNF maturation via elevated secretion of MMP9, as well as to altered synaptic plasticity in ex vivo hippocampal slices. As these findings link BDNF-modulated neuroplasticity to GR activation, we propose them as possible novel pathomechanism for stress-related disorders. This is supported by the fact that stress-related psychiatric disorders such as MDD and PTSD have been linked repetitively to BDNF, synaptic plasticity and dysregulation in GR signaling[31].

**Stress triggers secretory autophagy via FKBP51.** We found that GCs can trigger secretory autophagy, and that the stress-responsive protein FKBP51 acts as scaffolder and key driver of

this signaling pathway. The initial stressor, however, is at the crossroad between macroautophagy and secretory autophagy, and defines the cargo's fate. Our previous study showed that stress enhances macroautophagy via FKBP51[10]; here we show that high GC (dexamethasone) levels lead to lysosomal damage, both by observing enhanced GAL3/8 expression and via tfGal3 assays. These results indicate that intense or prolonged stress, which is well known for promoting disease progression of various psychiatric disorders such as PTSD, may lead to damage of lysosomal membranes, which in turn leads to an increase of secretory autophagy. In fact, we show that increased levels of lysosomal damage are paralleled by enhanced secretion of the established cargo CTSD.

Our mechanistic analyses of the dexamethasone-triggered activation of secretory autophagy confirmed that FKBP51 is a

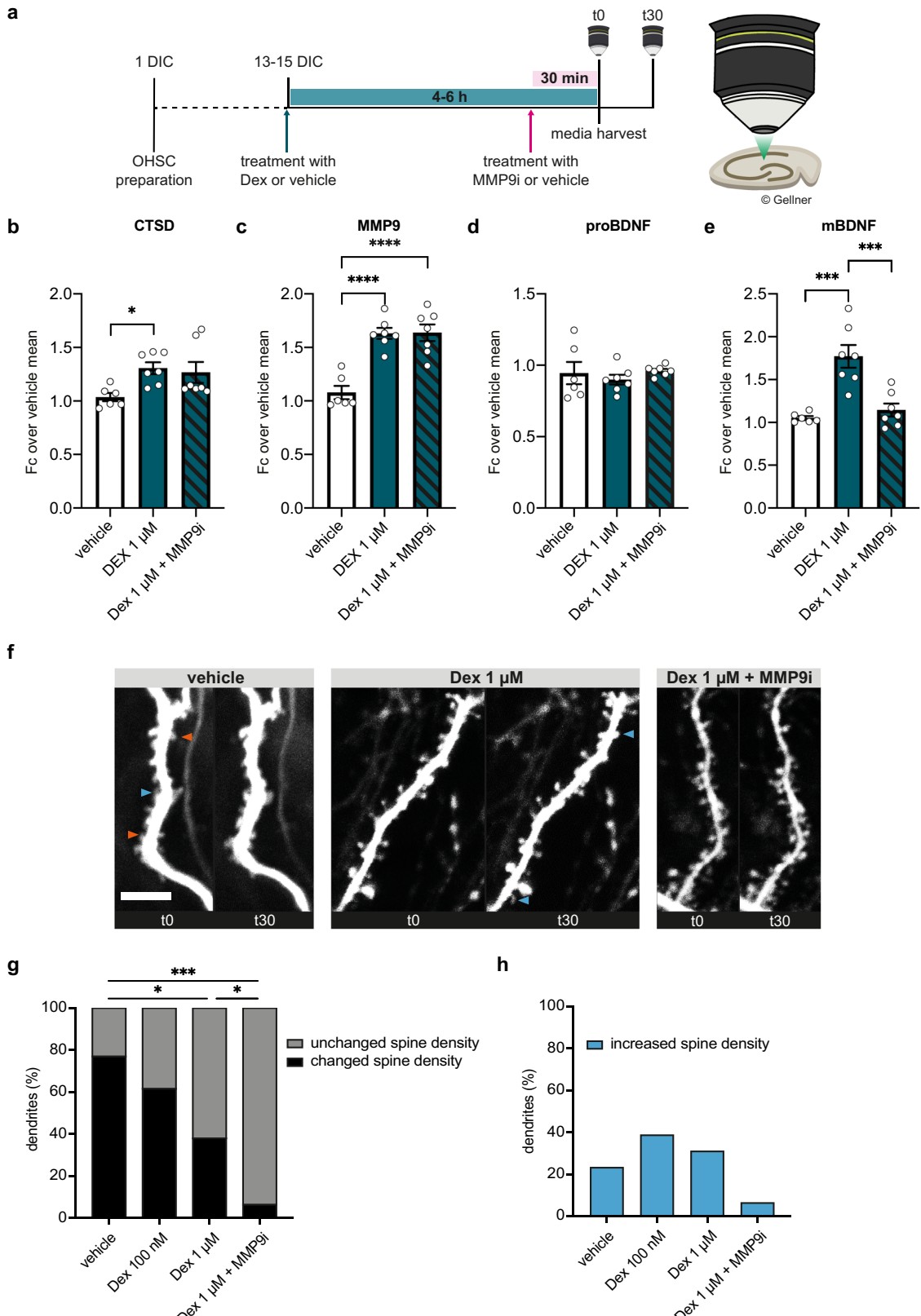

key component and essential driver of the last step of secretory autophagy. Here, FKBP51 acts by chaperoning the hetero-complex assembly of the vesicular R-SNARE SEC22B with the membranous Q-SNAREs thereby enabling vesicle-membrane fusion. Furthermore, we observed an increased and FKBP51-dependent interaction between the RQ-SNARE proteins and an enhanced secretion of CTSD upon dexamethasone treatment. Taken together, these results confirm that GC-mediated stress enhances secretory autophagy via FKBP51.

The analysis of the GC-induced secretome revealed that a surprisingly high proportion of secreted proteins (63%) show an autophagy-dependent release as indicated by a higher level of

**Fig. 6 Acute stress effects on hippocampal spine dynamics are mediated by MMP9-dependent BDNF maturation ex vivo. a** Experimental timeline of OHSC preparation, treatment and time lapse two-photon imaging of CA1. After 13–15 days in culture (DIC), OHSC are treated for 4-6 h with 1 µM Dex or vehicle and during the last 30 min additionally with 50 nM MMP9i or vehicle. After the treatment has ended, media is harvested for molecular analysis and dendrites of pyramidal neurons are imaged twice (t0 and t30 min) for capturing spine dynamics. Quantification of western blotting for **b** CTSD, **c** MMP9, **d** proBDNF, **e** mBDNF from harvested OHSC medium. Data shown as mean ± s.e.m. of $n = 3$ biologically independent samples. One-Way ANOVA with Tukey's multiple comparisons test was performed. *$P < 0.05$; ***$P < 0.001$; ****$P < 0.0001$. **f** Representative time lapse images of GFP-expressing dendritic segments in hippocampal region CA1 treated with either vehicle, 1 µM Dex and Dex 1 µM + 50 nM MMP9i (scale bar 5 µm). Orange arrowheads indicate disappearing spines, while blue arrowheads indicate novel spines appearing between t0 and t30. **g** Quantification and Chi-square analyses of dendritic spines classified into changed and unchanged between t0 and t30. **h** Quantitative representation of increasing spine densities only in different conditions, expressed as percentage of total spines counted.

secretion in WT (2-fold or more) compared to *Atg5* KO cells. The subsequent GO analysis of this group of secreted proteins (450) showed a high functional heterogeneity, suggesting that secretory autophagy is not specific for the regulation of a particular group of proteins. Enrichments (FDR < 0.01) in pathways of autophagy, cellular response to external stimuli and metabolism of proteins were in line with our expectations, given the high association of such functions with secretory autophagy and stress. Additionally, proteins related to the immune system were found to be enriched (Reactome analysis, FDR = 1.91E−8), which is in line with previous evidence showing the role of secretory autophagy in extracellular signaling of the immune response[12,14,15,32]. The enrichment of proteins involved in cell cycle, DNA replication, metabolism of RNA and programmed cell death could reflect the cellular response and adaption to stress. The successive neuron-specific analysis performed with SynGO revealed that 72 of the identified proteins exert synaptic functions and almost half of them (33) are involved in synaptic organization. This result supported our assumption that stress-induced secretory autophagy might have an effect on synaptic reorganization and neuroplasticity. In addition, the literature mining highlighted 163 proteins that have been found to be related to neuroplasticity in at least one peer-reviewed publication. A deeper analysis of the top candidates further illustrated that a large number of the neuroplasticity-related proteins is associated with psychiatric and neurodegenerative disorders (Table 1). This strengthened our supposition that stress might affect proteostasis not only in neurodegenerative disorders, but also in other, non-neurodegenerative psychiatric diseases, as proposed recently by Bradshaw and Korth[33].

Based on our results and previous publications, we propose a dual-step-stress model in which a mild, acute stress activates a first defense line that triggers lytic autophagy that, in turn, guarantees macromolecule recycling and degradation of potentially damaging aggregates. However, in the presence of prolonged or excessive stress, the cell switches to a second line of defense that leads to lysosomal damage and conveys the disposal of potentially damaging proteins to a secretory pathway, i.e., secretory autophagy. Furthermore, the finding that a large proportion of here-identified regulated proteins is involved in the immune response suggests that an initially positive effect increasing neuroplasticity and cognitive arousal, could come with the cost of neuroinflammation in the case of prolonged or chronic stress (see outline of proposed model in Fig. 7).

**MMP9 and BDNF—early proteins of the stress response?**
Among the identified proteins that are secreted in a stress- and autophagy-dependent manner, MMP9 stands out as key regulator in CNS development and plasticity[34]. As an endopeptidase, MMP9 cleaves—among other substrates—extracellular components, cell adhesion molecules, cell surface receptors, and neurotrophin precursors, such as proBDNF. Hence, MMP9 is implicated in active dendritic spine remodeling and stabilization,

pre- and postsynaptic receptor dynamics, and synaptic pruning[35,36]. In our study, we describe secretory autophagy as a novel stress-inducible mechanism for MMP9 secretion both in vitro and in vivo, elicited respectively by dexamethasone treatment or FS, leading to enhanced extracellular maturation of BDNF and to altered dendritic spine density ex vivo.

Elevated GC concentrations caused by prolonged stress may lead to structural changes and neuronal damage in distinct structures of the brain thereby possibly contributing to the development and progression of neuropsychiatric disorders[37]. Furthermore, accumulating evidence suggests that neuroplasticity, a central mechanism of neuronal adaptation, is altered in stress-related disorders and animal models of stress. Alongside its classical functions as a key neurotrophin during the development of the nervous system, BDNF modulates synaptic plasticity and is involved in stress-induced hippocampal adaptation and pathogenesis of MDD[38]. Our results demonstrate that secretory autophagy-mediated elevation in extracellular MMP9 levels causes an increase in the extracellular mBDNF/proBDNF ratio, a finding that has also been observed in MDD patients[39]. Thus, MMP9 might represent a mean to increase the availability of mBDNF, "optimizing" BDNF levels to facilitate synaptic plasticity and remodeling under prolonged stress conditions. Interestingly, it has been shown that stimulation of primary cortical neurons with BDNF can upregulate MMP9 mRNA and protein levels suggesting a positive feed-forward mechanism for BDNF maturation[40]. In line with these observations, Niculescu et al., have elegantly reported that MMP9-mediated mBDNF increase promotes synaptic clustering, the co-arrangement of highly synchronous synapses with similar activity into short dendritic regions enabling high precision synaptic inputs on hippocampal and cortical neurons[41–43]. Conversely, proBDNF exerts an antagonistic effect, promoting out-of-synch synaptic depression[44] and synapse retraction[45]. In line with this knowledge, our two-photon imaging analyses of OHSC show that a dexamethasone-induced and MMP9-dependent increase in extracellular mBDNF leads to an increased spine stabilization (unchanged spine density), but also to an increase in spine formation compared to vehicle. Interestingly, upon addition of MMP9i at the end of the "stress" period, we observe an even stronger stabilization of spine density with a complete absence of spine decrease and an overall reduction of novel spine formation below baseline levels. These data suggest that stress induces (1) an MMP9-dependent increase of extracellular mBDNF that leads to an increased spine formation, which is a well-known effect of mBDNF, and (2) an MMP9-independent counteraction of the spine reduction that occurs at baseline, and that might be enhanced by the imaging procedure (shock in the perfusion chamber, effect of DMSO[46]). These two effects lead to a net increase and stabilization in spine density that would possibly allow for an increased cognitive vigilance (and thus possibly also for resilience to further neuronal insults), making it seem plausible that lowered BDNF levels induce a state of increased vulnerability to stress and stress-

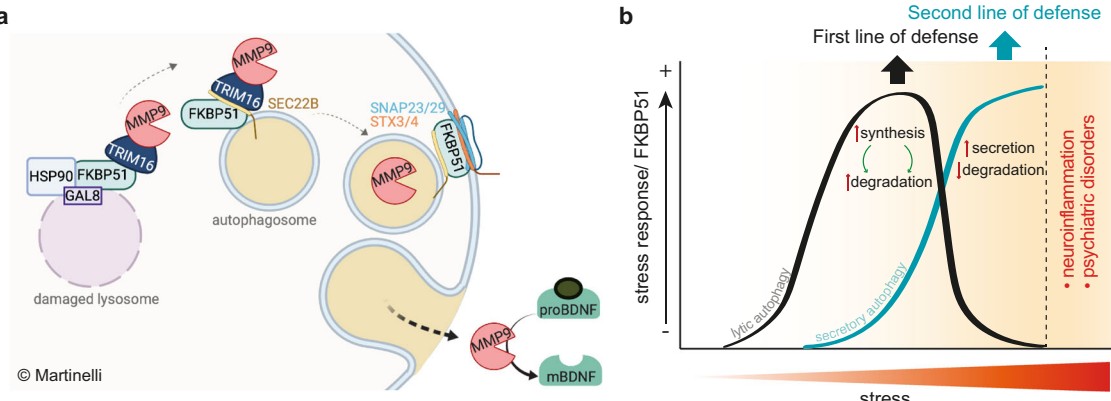

**Fig. 7 Proposed model based on the findings. a** Glucocorticoid-mediated stress induces lysosomal damage, which leads to the association of FKBP51 with the cargo receptor TRIM16 and the cargo protein (e.g., MMP9). This complex is transported on the autophagosome membrane via the interaction of FKBP51 with SEC22B. The cargo protein is internalized into the autophagosome which is transported to the plasma membrane. The vesicle-plasma membrane fusion is mediated via the SNARE-protein complex assembly, which is regulated by FKBP51 and sensitive to glucocorticoids. The membrane fusion leads to the release of the cargo proteins into the extracellular milieu. The increased secretion of extracellular MMP9 induces the cleavage of proBDNF to its mature form, which becomes the prevailing form in the extracellular space; **b** A first response to stress triggers lytic autophagy[10]. In case the stress persists, a second defense line is activated which switches the stress response from lytic to secretory autophagy. If stress further persists (e.g. chronic stress), the initially beneficial proteins secreted in response to stress, might lead to neuroinflammation and psychiatric disorders.

related disorders such as MDD, given the assessed importance of BDNF in relation to psychiatric disorders[47]. Indeed, numerous studies have described central and peripheral reductions in BDNF levels following chronic stress and in patients with MDD, indicating that optimum levels of mBDNF/proBDNF ratios are critical to ensure proper neuronal functioning[48]. Of note, having observed a rescuing effect of SAFit1 on the stress-induced increase of extracellular mBDNF in vivo, FKBP51 could be a promising target for the prevention of prolonged effects of an acute stress, such as a traumatic event. In fact, BDNF levels tend to normalize in response to several therapeutical interventions such as antidepressants[49] or physical activity[50].

Taken together, our findings show that GC-mediated stress triggers secretory autophagy thereby leading to an enhanced release of numerous proteins, of which many are involved in the immune response. Among them, we found MMP9 whose secretion results in an effective cleavage of proBDNF and accumulation of ready to act mBDNF in the extracellular space herewith enhancing synaptic stability and spine density ex vivo. We propose a model in which GC-mediated stress activates a fast molecular stress response that enhances neuroplasticity and cognition and might be followed, however, by a possible increase in neuroinflammation in case of prolonged or chronic stress (model summarized in Fig. 7). The antagonizing effect of SAFit1 is therefore of particular interest as a promising pharmacological tool to tackle stress-related psychiatric and neuroinflammatory diseases.

## Methods

**Cell culture**. The human cell lines HEK and SH-SY5Y were cultured at 37 °C, 6% CO₂ in Dulbecco's Modified Eagle Medium (DMEM) high glucose with GlutaMAX (Thermo Fisher, 31331-028), supplemented with 10% fetal bovine serum (Thermo Fisher, 10270-106) and 1% antibiotic-antimycotic (Thermo Fisher, 15240-062).

The murine microglia cell line SIM-A9 was cultured at 37 °C, 6% CO₂ in Dulbecco's Modified Eagle Medium (DMEM) high glucose with GlutaMAX (Thermo Fisher, 10566016), supplemented with 10% fetal bovine serum (Thermo Fisher, 10270-106), 5% horse serum (Thermo Fisher, 16050-122) and 1% antibiotic-antimycotic (Thermo Fisher, 15240-062).

**Treatments**. Dexamethasone (SIGMA) in DMSO was used at the specified concentrations. Bafilomycin (AlphaAesar) was used at the concentration of 100 nM in DMSO, SaFit1 was used at the concentration of 600 nM in DMSO, LLOMe was used at the concentration of 0.25 mM (SIGMA) in PBS. DMSO (SIGMA) was used

at a maximal final concentration of 1:10,000. The respective solvents are defined as vehicle throughout the manuscript.

**Transfections**. With 1× trypsin-EDTA (gibco, 15400-054) detached HEK, SH-SY5Y or SIM-A9 cells (2 × 10⁶) were resuspended in 100 µl of transfection buffer [50 mM Hepes (pH 7.3), 90 mM NaCl, 5 mM KCl, and 0.15 mM CaCl₂]. Up to 2 µg of plasmid DNA was added to the cell suspension, and electroporation was carried out using the Amaxa 2B-Nucleofector system (Lonza). Cells were replated at a density of 10⁵ cells/cm².

*Plasmids used*. FKBP51-FLAG as described in Wochnik et al. 2005[51]. GFP-SEC22B and Trim16-FLAG were kind gifts by Dr. Vojo Deretic (University of New Mexico).

**CRISPR-Cas9 KO generation**. Generation of SIM-A9 *Atg5* KO cell line: using Lipofectamine 3000 transfection reagent (Thermo Fisher Scientific, L3000001), cells were transfected with CRISPR-Cas9 *Atg5* plasmid. 48 h after transfection 2 µg/ml of puromycin (InvivoGen, ant-pr-1) was added to the medium. After 36 h the medium was changed, and single clones were manually picked and replated in single wells for expansion. KO clones were selected via western blot analysis.

SIM-A9 *Sec22b* and *Fkbp5* KO cell lines were generated using the Alt-R CRISPR-Cas9 System from Integrated DNA Technologies (IDT) according to the manufacturer's protocol. Briefly, RNA oligos (Supplementary Table 6; Alt-R CRISPR-Cas9 crRNA targeting murine *Sec22b* or *Fkbp5* and Alt-R CRISPR-Cas9 tracrRNA) were mixed in nuclease-free duplex buffer (IDT) in equimolar concentrations to generate a final duplex of 1 µM, heated at 95 °C for 5 min and combined with 1 µM Alt-R S.p. HiFi Cas9 Nuclease V3 diluted in Opti-MEM (Thermo Fisher Scientific). RNP complexes were assembled at RT for 5 min and mixed with Lipofectamine RNAiMAX reagent (Thermo Fisher Scientific) and Opti-MEM (Thermo Fisher Scientific) and incubated for 20 min at RT to form transfection complexes. Subsequently, 40,000 SIM-A9 cells/well were reverse-transfected using complete culture media without antibiotics in a 96-well tissue culture plate with a final RNP concentration of 10 nM. After 48 h (37 °C, 5% CO₂) single cell clones were obtained by array dilution method, expanded, and analyzed by western blotting.

For the generation of *FKBP5* KO SH-SY5Y cell line, cells were co-transfected with a pool of three CRISPR/Cas9 plasmids containing gRNA targeting human *FKBP5* and a GFP reporter (Santa Cruz, sc-401560) together with a homology-directed repair plasmid (sc-401560-HDR) consisting of three plasmids, each containing a homology-directed DNA repair (HDR) template corresponding to the cut sites generated by the FKBP51 CRISPR/Cas9 KO Plasmid (sc-401560). 48 h after transfection, medium was changed to 2 µg/ml puromycin-containing medium. After 36 h, single clones were manually picked and replated in single wells for expansion. KO clones were selected via western blot analyses.

*Plasmids used*. FKBP51 CRISPR/Cas9 KO Plasmid (h), Santa Cruz #sc-401560. pSpCas9 BB-2A-Puro (PX459). V2.0 vector containing Atg5-targeting gRNA (Atg5 [house mouse] CRISPR gRNA 2: TATCCCCTTTAGAATATATC) purchased from GenScript.

**Western blot analysis**. Protein extracts were obtained by lysing cells in 62.5 mM Tris, 2% SDS, and 10% sucrose, supplemented with protease (Sigma, P2714) and phosphatase (Roche, 04906837001) inhibitor cocktails. Samples were sonicated and heated at 95 °C for 10 min. Proteins were separated by SDS-PAGE and electro-transferred onto nitrocellulose membranes. Blots were placed in Tris-buffered saline solution supplemented with 0.05% Tween (Sigma, P2287) (TBS-T) and 5% nonfat milk for 1 h at room temperature and then incubated with primary antibody (diluted in TBS-T) overnight at 4 °C. Subsequently, blots were washed and probed with the respective horseradish-peroxidase- or fluorophore-conjugated secondary antibody for 1 h at room temperature. The immuno-reactive bands were visualized either using ECL detection reagent (Millipore, WBKL0500) or directly by excitation of the respective fluorophore. Recording of the band intensities was performed with the ChemiDoc MP system from Bio-Rad.

*Quantification*. All protein data were normalized to Actin, GAPDH or Vinculin, which were detected on the same blot.

*Primary antibodies used (also for Wes, ProteinSimple)*. FLAG (1:7000, Rockland, 600-401-383), FKBP51 (1:1000, Bethyl, A301-430A), FKBP5 (D5G2) (1:1000, Cell Signaling, #12210), Actin (1:5000, Santa Cruz Biotechnology, sc-1616), GAPDH (1:8000, Millipore CB1001), TRIM16 (1:1000, Bethyl A301-160A), CTSD (for Human) (1:50, Abcam, ab6313), CTSD (for Mouse) (1:50, Abcam, ab207549), SNAP29 (1:1000, Sigma SAB1408650), SNAP23 (1:1000, Sigma, SAB2102251), STX3 (1:1000, Sigma, SAB2701366), STX4 (1:1000, Cell Signaling, #67657), GAL8 (1:1000, Santa Cruz, sc-28254), GAL3 (1:1000, Santa Cruz, sc-32790), SEC22B (1:1000, Abcam, ab181076 and 1:1000, Thermo Fisher, OSS00040W), Vinculin (1:1000, Cell Signaling, #13901)

**Co-immunoprecipitation**. In case of FLAG- or GFP-tag IP, cells were cultured for 3 days after transfection. Cells were lysed in co-IP buffer [20 mM Tris-HCl (pH 8.0), 100 mM NaCl, 1 mM EDTA, and 0.5% Igepal complemented with protease (Sigma) and phosphatase (Roche, 04906837001) inhibitor cocktail] for 20 min at 4 °C with constant mixing. The lysates were cleared by centrifugation, and the protein concentration was determined and adjusted (1.2 mg/ml); 1 ml of lysate was incubated with 2.5 µg of FLAG or GFP antibody overnight at 4 °C with constant rotating. Subsequently, 20 µl of bovine serum albumin (BSA)–blocked protein G Dynabeads (Invitrogen, 100-03D) were added to the lysate-antibody mix followed by a 3-h incubation at 4 °C. Beads were washed three times with PBS, and bound proteins were eluted with 100 µl of 1 × FLAG peptide solution (100–200 µg/ml, Sigma F3290) in PBS for 30 min at 4 °C. In case of precipitation of GFP tag, elution was performed by adding 60 µl of Laemmli sample buffer and by incubation at 95 °C for 5 min. 5–15 µg of the input lysates or 2.5–5 µl of the immunoprecipitates were separated by SDS–polyacrylamide gel electrophoresis (PAGE) and analyzed by western blotting. When quantifying co-immunoprecipitated proteins, their signals were normalized to input protein and to the precipitated interactor protein.

**Tandem fluorescent-tagged galectin-3 (tfGal3) assay**. SH-SY5Y cells were transfected with a the ptfGalectin3 plasmid (Addgene.org, #64149) that expresses Galectin3 tagged with both EGFP (inactivated in lysosomes) and mRFP (resists inactivation in lysosomes)[20]. Transfection in SH-SY5Y cells was performed using Lipofectamine 3000 transfection reagent (Thermo Fisher Scientific, L3000001), followed by drug treatment the next day. Cells were fixed (1% PFA for 1 h) 1 day later and analyzed by laser scanning microscopy (Leica Confocal Sp8). Vesicles were counted by an experimenter blind to the conditions.

**Quantitative PCR (RT-qPCR) analysis**. Total RNA was isolated from SIMA-A9 cells using the RNeasy Mini Kit (Qiagen, 74104). 5 µg of total RNA were reverse transcribed using a High Capacity cDNA Reverse Transcription Kit (Thermo Fisher, 4368814). Quantitative PCRs were performed using the TaqMan StepOnePlus Real-Time PCR System and a TaqMan 5′-nuclease probe method (Thermo Fisher). All transcripts were normalized to *Hprt* and *Gapdh*. Predesigned human TaqMan assays (Thermo Fisher) were used for quantifying gene expression of *Bdnf* (Mm04230607_s1), *Hprt* (Mm03024075_m1), and *Gapdh* (Mm99999915_g1).

**ELISA**. The solid-phase sandwich ELISA was performed according to the manufacturer protocol for the detection of the following proteins: CTSD (abcam ab213845), MMP9 (abnova KA0398), proBDNF (AssaySolutions AYQ-E10246), and mBDNF (AssaySolutions AYQ-E10225). Briefly, microwells were coated with the antibody of interest followed by a first incubation with biotin-coupled secondary antibody, a second incubation with streptavidin-HRP and a final incubation with the SIM-A9 culture medium. Amounts of secreted proteins were detected with a plate reader (iMARK, Bio-Rad) at 450 nm.

**Pull-down**. The pull-down was performed according to the manufacturer's manual (MagneGST TM Pull-Down System; Promega, V8870). As bait protein, 200 ng of human recombinant TRIM16 (Abnova, H00010626-P01) was immobilized to magnetic beads. Purified FKBP51 (110 ng) was used for the binding reaction. The binding reaction was carried out in a final volume of 400 µl of binding buffer

supplied. FKBP51 was purified as described before Gassen et al.[52]. Input material and eluates were analyzed using western blotting.

**Animal housing conditions**. C57BL/6 male mice (Martinsried, Germany), *Fkbp5*-KO and respective WT males (Martinsried, Germany) at 8–12 weeks of age were housed in cages of 4–5 at 21 °C, 50% humidity ±10% with a 12:12 h light-dark cycle with food and water ad libitum before experiments. Allocation of animals to experimental groups in respect to date of birth and litter was done using random number generation approach (RAN function in Excel). All procedures were done in accordance with European Communities Council Directive 2010/63/EU and approved by Government of Upper Bavaria.

**In vivo brain microdialysis in mice**
*Surgeries*. Surgeries were performed as described previously in Anderzhanova et al.[53] and in Anderzhanova et al.[54] under 2% isoflurane in oxygen (Abbot, India), Metacam® 0.5 mg/kg, s.c (Boehringer Ingelheim GmbH, Germany), and Novalgine 200 mg/kg, s.c. (Sanofi-Aventis, Germany) systemic anesthesia and analgesia. Lidocaine 2% (Xylocaine, AstraZeneca) was used for local anesthesia. Coordinates for microdialysis probe guide cannula implantations into the right mPFC were set in accordance to Paxinos and Franklin Mouse Brain Atlas[55] with bregma as a reference point: AP 2.00 mm, ML 0.35 mm, and DV − 1.50 mm. Stereotaxic manipulations were performed in the TSE stereotaxic frame (TSE Systems Inc., Germany). Animals were allowed for 7 days of recovery in individual microdialysis cages ($16 \times 16 \times 32$ cm$^3$). Metacam (0.5 mg/kg, s.c) was injected within the first 3 days after surgeries, when required.

*Microdialysis*. The perfusion setup was the line comprised of FET tubing of 0.15 mm ID (Microbiotech Se, Sweden), a 15 cm-PVC inset tubing (0.19 mm ID), a dual-channel liquid swivel (Microbiotech Se, Sweden). Perfusion medium was sterile RNase free Ringer's solution (BooScientific, USA) containing 1% BSA (Sigma-Aldrich, Cat.N A9418). Perfusion medium was delivered to the probe at the flow rate of 0.38 µl/min with the syringe pump (Harvard Apparatus, USA) and withdrawn with the peristaltic pump MP2 (Elemental Scientific, USA) at the flow rate of 0.4 µl/min. Microdialysis CMA 12 HighCO Metal Free Probe was of 2 mm length membrane with 100 kDa cut off (Cat.N. 8011222, CMA Microdialysis, Sweden). All lines were treated with 5% polyethylenimine for 16 h and then with H2O for 24 h before experiments. The microdialysis probe was inserted into the implanted guide cannula (under 1–1.5 min isoflurane anesthesia, 2% in air) 6 days after the stereotaxic surgery and 18 h before the samples collection. A baseline sample collection phase (three samples) was always preceding the FS, which allowed us to express the changes in the extracellular content of proteins as relative to the baseline values. On the experimental day, microdialysis fractions were constantly collected (for 30 min) into Protein LoBind tubes (Eppendorf, Germany) at a perfusion rate of 0.4 µl/min. During collection time, tubes were kept on ice. After collection of three baseline samples animals were transferred to the foot shock (FS) chamber (ENV-407, ENV-307 A; MED Associates, 7 St Albans, VT, USA) connected to constant electric flow generator (ENV-414; MED Associates) and a FS (1.5 mA × 1 s × 2) was delivered. After this procedure, mice were returned to the microdialysis cage where two post-FS samples were collected. To examine an effect of ULK1 inhibitor MRT 68921 on stress-evoked changes in extracellular content of proteins, the drug was injected intraperitoneally in a dose of 5.0 mg/kg and in a volume 10 ml/kg 4 h before application of FS (the drug was prepared freshly dissolving a stock solution [60%EthOH/40% DMSO mixture] with saline in proportion 1:20). 30 min microdialysis fractions were collected on ice into 1.5 ml protein LoBind tubes (Eppendorf, Germany) preloaded with 0.5 µl protease inhibitor cocktail 1:50 (Roche) and then immediately frozen on dry ice. At the end of the experiment, probes were removed, brains were frozen and kept at −80 °C for the probe placement verification. 40 µm brain sections were stained with cresyl violet (Carl Roth GmbH, Germany) and probe placement was verified under a microscope using Paxinos and Franklin mouse atlas[55]. If probe placement was found to be out of the targeted region of interest, the respective samples were excluded from the study.

**Time-lapse two-photon imaging of organotypic hippocampal slice cultures**
*Preparation of OHSC*. Neonatal Thy1-GFP M mice with a sparse expression of green fluorescent protein (GFP) in principal neurons in cortex and hippocampus (Jackson Laboratory Stock #007788) were used for all experiments. Pups aged between P7-9 (sex not determined) were decapitated, brains removed, and hippocampi isolated from both hemispheres in ice-cold 1× minimum essential medium (MEM) with EBSS, 25 mM HEPES and 10 mM Tris buffer (pH 7.2) supplemented with Penicillin (100 I.U./ml) and Streptomycin (100 µg/ml). Hippocampi were cut into coronal slices (thickness 350 µm) using a tissue chopper (McIlwain). Slices were transferred onto "confettis" Millipore biopore membrane, ~3 × 5 mm) which were placed on semiporous Millicell-CM inserts (0.4 µm pore size; Merck-Millipore). The inserts were put into cell culture dishes (35 mm, 4 slices/dish). OHSC were cultured according to the interface method in 1 ml medium per dish at pH 7.2, 35 °C and a humidified atmosphere with 5% CO2. Culture medium contained 0.5 × MEM with EBSS and 25 mM HEPES, 1mM L-Glutamine, 25% Hanks' Balanced Salt solution, 25% heat-inactivated horse serum,

Penicillin (100 I.U./ml) and Streptomycin (100 µg/ml; all Fisher Scientific). The medium was changed 1 day after preparation and every other day afterwards. Imaging experiments were performed between 13 and 15 days in culture (DIC). On the day of the experiment, inserts were transferred to fresh culture medium containing either 100 nM or 1 µM dexamethasone or vehicle (DMSO). After 4–6 h, a confetti containing one OHSC was carefully transferred to the interface chamber under the microscope objective superfused with carbogenated artificial cerebrospinal fluid containing (in mM): NaCl 130, KCl 2.75, MgSO₄ 1.43, NaH₂PO₄ 1.1, NaHCO₃ 28.82, CaCl₂ 2.5, MgCl₂ 1, HEPES 2.5 and glucose 11, pH7.4. OHSCs were acclimatized for 15–20 min at 35 °C before time-lapse imaging commenced. In inhibitor experiments, 50 nM of MMP9-Inhibitor I were added to the culture medium 30 min before transfer to the microscope. Medium was harvested and snap frozen in liquid nitrogen directly after the OHSC used for imaging had been removed from the cell culture dish.

*Two-photon imaging of OHSC.* A custom built two-photon microscope was controlled by ScanImage Software (Vidrio) and driven with a Ti:sapphire Laser (Chameleon Vision-S, Coherent) tuned to 910 nm for GFP excitation. Fields of view containing GFP positive dendritic segments from multiple different neurons originating in the CA1 region of the culture slice were imaged using a ×40 water immersion objective (Olympus). Time lapse stacks (1024 × 1024 pixels, xy 0.08 µm/ px, z 0.8 µm, frame averaging factor 4, frame rate 0.41 Hz) from the region of interest were recorded at baseline (0 min) and after 30 min.

*Two-photon image analysis.* A custom MATLAB® script was used to count and track the fate of individual spines on 3–5 individual dendritic segments (length >30 µm) in each imaged OHSC at 0 and 30 min (t0, t30). Spine density was calculated for each slice as the total number of spines per total length of the analyzed dendritic segments for each time point. Spine density at t30 was then and normalized to baseline (0 min) at t0, indicating the spine density change then used for statistical analysis.

*Statistics.* Three different OHSC preparations were used and conditions were randomly distributed between cultures from individual pups. Spine density change calculated as described in the previous section was transformed to categorial classes (increased/decreased/unchanged spine density) to enable analysis of responsiveness of spine dynamics to the different treatments by Chi-squared statistics.

### Interactome analyses
*Sample preparation.* All samples were prepared in quadruplicate. HEK cells were transfected with a FLAG-tagged FKBP51 expressing plasmid or a FLAG expressing control vector. 48 h after transfection, a FLAG IP was performed on all the samples and the eluted proteins were separated by SDS gel electrophoresis. Separated proteins were stained with Coomassie staining solution for 20 min and destained over night with Coomassie destaining solution. Each gel lane was cut into 21 ~2.5 mm slices per biological replicate and these were further cut into smaller gel pieces.

*In-gel-trypsin digestion and peptide extraction.* The gel pieces were covered with 100 µl of 25 mM NH₄HCO₃/50% acetonitrile in order to destain the gel pieces completely and mixed for 10 min at room temperature. The supernatant was discarded, and this step was repeated twice. Proteins inside the gel pieces were reduced with 75 µl of 100 mM dithiothreitol/25 mM NH₄HCO₃ mixed at 56 °C for 30 min in the dark. The supernatant was discarded and 100 µl of 200 mM iodoacetamide were added for alkylation to the gel pieces and mixed for 30 min at room temperature. The supernatant was discarded, and the gel pieces were washed twice with 100 µl 25 mM Na₄HCO₃/50% acetonitrile and mixed for 10 min at room temperature. The supernatant was discarded, and gel pieces were dried for ~20 min at room temperature. Proteins were digested with 50 µl trypsin solution [5 ng/µl trypsin/25 mM NH₄HCO₃] over night at 37 °C. Peptides were extracted from the gel pieces by mixing them in 50 µl of 2% formic acid/50% acetonitrile for 20 min at 37 °C and subsequently sonicating them for 5 min. This step was repeated twice with 50 µl of 1% formic acid/50% acetonitrile. The supernatants of three slides were pooled. Each sample generated 7 samples for submission to LC-MS/MS analysis.

*LC MS/MS.* Tryptic peptides were then dissolved in 0.1% formic acid and analyzed with a nanoflow HPLC-2D system (Eksigent, Dublin, CA, USA) coupled on-line to an LTQ-Orbitrap XL™ mass spectrometer. Samples were on-line desalted for 10 min with 0.1% formic acid at 3 µl/min (Zorbax-C18 (5 µm) guard column, 300 µm × 5 mm; Agilent Technologies, Santa Clara, CA, USA) and separated via RP-C18 (Dr. Maisch, Germany, 3 µm) chromatography (in-house packed Pico-frit column, 75 µm × 15 cm, New Objective, Woburn, MA, USA). Peptides were eluted with a gradient of 95% acetonitrile/0.1% formic acid from 10 to 45% over 93 min at a flow rate of 200 nl/min. Column effluents were directly infused into the mass spectrometer via a nano-electrospray ion source (Thermo Fisher Scientific). The mass spectrometer was operated in positive mode applying a data-dependent scan switch between MS and MS/MS acquisition. Full scans were recorded in the Orbitrap mass analyzer (profile mode, m/z 380–1600, resolution R = 60,000 at m/z 400). The MS/MS analyses of the five most intense peptide ions for each scan were recorded in the LTQ mass analyzer in centroid mode.

*Peptides and proteins identification.* The MS raw data were first converted in a peaks list using Bioworks v. 3.3.1 (Thermo Fischer Scientific, San Jose, CA) and then were searched against the SwissProt_15.3 (uniprot 29.05.09) human database using Mascot search algorithm (v. 2.2.07, www.matrix.com).

The precursor and fragment mass tolerance were set to 20 ppm and 0.8 Da, respectively. Trypsin/P including one missed cleavage was set as enzyme. Methionine oxidation and carbamidomethylation of cysteine were searched as variable and static modifications, respectively. The significance level in protein identification was set p > 0.95, the peptide score cut-off was set 20. Proteins identified with at least three peptides were considered for the protein data processing.

The identified proteins overlapping in the four replicates were considered as protein interactors.

The mass spectrometry proteomics data have been deposited to the ProteomeXchange Consortium via the PRIDE[56] partner repository with the dataset identifier PXD017328 and 10.6019/PXD017328

### Secretome analyses
*Sample preparation.* All samples were prepared in triplicates. For each replicate 2 × 150 mm culture dishes, containing each 20 ml of medium, were used. WT and *Atg5* KO SIM-A9 cells were cultured in methionine-free medium [DMEM, high glucose, no glutamine, no methionine, no cystine (Thermo Fisher, 21013024), 2 mM L-Glutamine (Thermo Fisher, 25030081), 1 mM sodium pyruvate (Thermo Fisher, 11360070), 0.2 mM L-Cysteine (Sigma, C7352), 10% dialyzed fetal bovine serum (Thermo Fisher, 30067334), 1% Antibiotic-Antimycotic (Thermo Fisher, 15240-062)] for 30 min (12 h and 30 min before supernatant collection). After 30 min, medium was changed to AHA enriched medium [methionine-free medium with 20 µM Click-IT™ AHA (L-Azidohomoalanine) (Thermo Fisher, C10102)] to label newly synthesized proteins. After 6 h, cells were treated with 100 nM dexamethasone or DMSO (vehicle) at a dilution of 1:10,000 (6 h before supernatant collection). After 3 h (3 h before supernatant collection), all media were refreshed in order to analyze proteins that were secreted during the last 3-h only. After 3 h, supernatants were collected and centrifuged at 1000 × g for 5 min. EDTA-free proteinase inhibitor cocktail (Sigma, S8830) was added and supernatants were frozen at −20 °C. The next day, supernatants were thawed on ice and concentrated with Amicon Ultra 15 centrifugal filters, Ultracell 3 K (Millipore, UFC900324) at 4000 × g at 4 °C for 5 h. The concentrated samples were transferred into new 1.5 ml tubes (Sigma, Z606340). For the enrichment and digestion of AHA-labeled proteins, the Click-iT™ Protein Enrichment Kit (Thermo Fisher Scientific) was used according to the instructions of the supplier, despite that only half of the suggested volumes were used[22]. Resulting peptides were further desalted by using 0.1% formic acid in 50 % acetonitrile in the micro-column format (three discs, Ø 1.5 mm, C18 material, 3 M Empore per micro-column were used)[57]. After drying in a centrifugal evaporator, the samples were stored at −20 °C until LC-MS/MS analysis.

*LC-MS/MS.* LC-MS/MS measurement of peptides in eluates was performed using a nanoLC UltiMate 3000 (Thermo Fisher Scientific) coupled to a quadrupole-Orbitrap Q Exactive HF-X mass spectrometer (Thermo Fisher Scientific). Peptides separated on an Acclaim PepMap analytical column (0.1 mm × 15 cm, C18, 2 µM, 100 Å; Thermo Fisher Scientific) using a 60 min linear gradient from 3 to 28% solvent B [0.1% formic acid, 5% DMSO in acetonitrile] in solvent A [0. 1% formic acid, 5% DMSO in water] at a flow rate of 10 µL/min. The mass spectrometer was operated in data-dependent acquisition and positive ionization mode. MS1 spectra were acquired over a range of 360–1300 m/z at a resolution of 60,000 in the Orbitrap by applying an automatic gain control (AGC) of 3e6 or maximum injection time of 50 ms. Up to 12 peptide precursors were selected for fragmentation by higher energy collision-induced dissociation (HCD; 1.3 m/z isolation window, AGC value of 1e5, maximum injection time of 22 ms) using 28% normalized collision energy (NCE) and analyzed at a resolution of 15,000 in the Orbitrap.

*Peptide and protein identification and quantification.* Peptide and protein identification and quantification was performed using MaxQuant (version 1.6.0.16)[58] by searching the tandem MS data against all murine canonical and isoform protein sequences as annotated in the Swissprot reference database (25,175 entries, downloaded 13.07.2018) using the embedded search engine Andromeda[59]. Carbamidomethylated cysteine was set as fixed modification and oxidation of methionine and N-terminal protein acetylation as variable modification. Trypsin/P was specified as the proteolytic enzyme and up to two missed cleavage sites were allowed. Precursor tolerance was set to 4.5 ppm and fragment ion tolerance to 20 ppm. The minimum peptide length was set to seven and all data were adjusted to 1% PSM and 1% protein FDR. Intensity-based absolute quantification[60] was enabled within MaxQuant.

*Data analysis.* The Perseus software suite (v. 1.6.2.3)[61] was used to filter out contaminants, reverse hits, and protein groups, which were only identified by site. Only protein groups that were detected in at least two out of the three replicates in at least one condition were considered for analysis.

The filtered data was log normalized and missing values were imputed according to the normally distributed imputation algorithm implemented in the Perseus framework. Default values were used (width: 0.3; down shift: 1.8). To find the significantly regulated protein groups a multiple *t*-test (volcano plot) analysis was performed with a difference of s0 = 1 and a false discovery rate (FDR) of 0.01. Proteins with a significant (FDR < 0.01) and at least 2-fold (s0 > 1) increased secretion in the WT compared to the Atg5 KO samples were analyzed with the reactome pathway browser (www.reactome.org; pathway browser version 3.6, reactome database release 70)[62]. The same list was also analyzed with the SynGO knowledgebase (www.syngoportal.org)[23].

**Automated literature search**. Protein lists resulting from the interactome and secretome analyses were subjected to an automated literature search performed with a custom python code, revealing as output the number of references found for each search. For the interactome list, the algorithm performed an automated PubMed search of the protein with "autophagy", "proteostasis", or "ubiquitin proteasome system". For the secretome list, the algorithm performed an automated PubMed search of each protein with "neuroplasticity". False negatives resulting from protein name or abbreviation ambiguity were manually corrected.

**Statistics and reproducibility**. Except from proteomics data, all statistical analyses were performed with Prism version 8.1.1 (GraphPad Software, La Jolla California USA, www.graphpad.com). Data distribution was tested with the Shapiro–Wilk test. Individual data points are shown when possible. Error bars indicate standard error mean (s.e.m.). Analyses of paired measurements were performed using the Sidak's multiple comparisons test for individual comparisons and two-way ANOVA test for group comparisons. Unpaired analyses were performed using one-way unpaired *t*-test with appropriate correction depending on the distribution and variance: normally distributed data with different variance were corrected with Welch's correction; non-normally distributed data were analyzed with Mann–Whitney *U* test. For multiple comparisons of normally distributed data, Tukey's comparison test was applied, while for non-normally distributed data, the Kruskal–Wallis test was performed. $P < 0.05$ was considered statistically significant.

Each co-immunoprecipitation experiment was repeated independently at least 3 times with similar results.

**Reporting summary**. Further information on research design is available in the Nature Research Reporting Summary linked to this article.

## Data availability
The interactome mass spectrometry proteomics data have been deposited to the ProteomeXchange Consortium via the PRIDE[39] partner repository with the dataset identifier PXD017328. The secretome mass spectrometry proteomics data have been deposited to the ProteomeXchange Consortium via the PRIDE partner repository with the dataset identifier PXD017076. Source data are provided with this paper.

## Code availability
Scripts for automated literature search have been deposited on git hub in the following repository: [https://github.com/silviamartinelli/paper-scripts]. The custom MATLAB® script was adapted from the legacy script scim_SpineAnalysis in the open source ScanImage release r3.8, accessible via [http://scanimage.vidriotechnologies.com/]. The modified version is accessible from the author via reasonable request.

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

## Acknowledgements

Many thanks to Elisabeth Binder for her support and help with this study. This study was funded by a NARSAD Young Investigator Award by the Brain and Behavior Research Foundation, honored by P&S Fund (Awarded to N.C.G., Grant ID 25348).

## Author contributions

S.M., E.A.A. and N.C.G. designed the study and performed the experiments. S.M. and N.C.G. wrote the manuscript. S.M, E.A.A., A.P. and N.C.G. analyzed the data. S.W., F.D., K.W., G.M., C.W.T. and B.K. performed and analyzed the proteomic experiments. M.L.P., E.A.A. and M.V.S. performed mouse work. M.D., D.H., V.S. and A.K.G. performed two-photon microscopy and related experiments. T.M.G. and F.H. synthesized SAFit1 and purified proteins. K.H., T.B., T.E., S.V.S., U.S., J.H., M.L.P. and L.J. contributed to the experiments. N.C.G. supervised the study. All authors contributed to the final version of the manuscript.

## Funding

## Competing interests

The authors declare no competing interests.
