## [Peer Review File · Nature Communications]

Reviewers' comments:

Reviewer #1 (Remarks to the Author):

The paper by Martinelli Anderzhanova et al presents new insights in regulation of secretory autophagy, identifies new and important regulatory proteins of secretory autophagy. The study shows for the first time, how one important cargo protein of secretory autophagy, the matrix metalloproteinase 9 (MMP9) can contribute to more behavior-related, extracellular abundance of mature BDNF.

BDNF is one of the most important key proteins in synaptic plasticity, it can be stored in synapses and undergoes activity-dependent secretion. BDNF is involved in diverse synaptic processes, including synapse maturation, synapse refinement, synaptic transmission and even pre- and postsynaptic LTP. Two BDNF isoforms are known to regulate synaptic transmission, the mature BDNF, a homodimeric protein with high affinity to TrkB, and proBDNF, an isoform carrying the so called pro-domain. Even the cleaved pro-domain has been shown to be involved in synaptic processes. proBDNF shows high-affinity to p75, another neurotrophin receptor. There is an ongoing debate about proBDNF secretion and processing.

In this important study, the authors show that FKBP51, a stress responsive co-chaperone is critically involved in secretory autophagy. Notably, MMP9 is a cargo of these granules and undergoes regulated secretion, possibly for the local cleavage of proBDNF to mBDNF. The finding that cellular stress-related release from autophagosome-like granules/vesicles is associated with extracellular mBDNF abundance is credible.

The design of the study is straightforward and conceptually strong. The experiments are convincing and there is a lot of important, new and relevant information available. For instance, the interactome of FKBP51 is well worked out and the experiments help better understand how FKBP51-positive autophagosomes behave within the trafficking pathway. Much of the work has been done in cell lines (SH5YSY, neuroblastoma-like cell line; Hek293, SIM-A9, microglia-like), but the detailed information about key proteins in the secretory autophagosome (SEC22B, RACK1, UBC12) pathway will help to find out how secretory autophagy is acting in neurons, at synapses or between cell types (microglia) at synapses.

The in vivo experiments clearly show, with a new approach, direct determination of behavior-related secretion of BDNF isoforms. However, we do not learn from which cells BDNF is secreted, an aspect beyond the scope of this study. Nevertheless, the data convincingly reveal that MMP9 and BDNF appear in the prefrontal cortex in the course of a behavioral test and the data show that MMP9 contributes to more mBDNF.

I'm sure that these proof-of-principle experiments will motivate many researchers in the field to look again at the fundamental biology of synaptic BDNF in 'real' learning and memory paradigms.

The methodology is of highest quality. I think that this paper is of substantial importance. For me, there are some major concerns that have to be addressed before it can be considered for publication in Nature communication.

Major comments:

In general:

1) The authors write (line 88): ...an increase in cleavage of pro-brain-derived neurotrophic factor (proBDNF) to its mature form (mBDNF) both in vitro and in vivo.

However, the data do not show that proBDNF cleavage leads to less proBDNF and more mBDNF. The data show the same amount of proBDNF (in vitro-ELISA) and in vivo (Immunoblotting; but size is not given). If there is cleavage on cost of proBDNF, Western analysis should show less proBDNF at its full

size relative molecular weight and more mBDNF at 13 kDa. The interpretation and discussion of the data should be in line with the data.

2) Fig. 3: Secretome. proBDNF and mBDNF should appear in the secretome, but the authors do not show the data and do not discuss it (or I have overseen it). My question is: Is there more or less pro-domain or mBDNF in the secretome of SIM-A9 cells after Atg5KO or is there an explanation why there is not much BDNF in the secretome?

In line with the overall concept of the study, one assumption would be that if BDNF is in a different vesicle and is not co-released with MMP-9, one would expect to see more pro-domain in case of Dex/Atg5KO. Is this really extracellular cleavage on cost of proBDNF or are there other explanations?

3) Fig 4: Important data depend on SIM-A9 cells, a microglia-like cell line. The data suggest that the cells produce BDNF and secrete it. However, western analysis is missing. Western blotting data would help to find out whether proBDNF is intracellular and then secreted as proBDNF. It can well be that this is secretion or abundance of cleaved pro-domain versus mBDNF. In Western blotting after SDS-PAGE, mBDNF should appear at 13 kDa, the cleaved pro-domain should run close to 15 – maybe 20 kDa, the uncleaved pro-domain may be expected, in its glycosylated form at about 32-34 kDa. Immunoblotting from supernatants, maybe after IP or after protein concentration, should answer this important aspect. It may be that the authors find other or new anti-pro/anti-BDNF immunoreactive bands. This experiment can help to better understand the overall data set.

2) Fig. 5: The authors quantified BDNF and proBDNF in microdialysates. The method is rather new and of outstanding importance. The method section is not clear here. Antibodies or other reagents are not given. The method is immunoblotting-dependent (line 286). Original data (proBDNF, BDNF, MMP9) should be shown. The authors should cite or point to another study showing how this new approach was developed and verified.

3) In Fig. 6, after the in vivo data, the authors present the SAFit1 inhibition of FKBP51 in cell culture. The data fit better to Figure 3.

Again, as mentioned above, the data are not supported by Western blotting. Here, it is ELISA detection of proBDNF or mBDNF from supernatants. ProBDNF is unchanged, but there is less mBDNF after SAFit1 inhibition. It can well be that MMP9 is acting at a different place. Or another yet unknown MMP9-associated process leads to more secretion of mBDNF. This might explain that there is almost no change in proBDNF in Fig. 4, 5e and 5i. Fig. 6. Anyhow, in Fig. 5, it is very clear that mBDNF goes up, in vivo, extracellular, regulated and behavior-related. This direct verification of regulated, in vivo BDNF secretion is a very important finding.

Minor

1) The idea of functional extracellular cleavage of proBDNF is quite old. Please cite the corresponding landmark paper: Pang, P.T., H.K. Teng, E. Zaitsev, N.T. Woo, K. Sakata, S. Zhen, K.K. Teng, W.H. Yung, B.L. Hempstead, and B. Lu. 2004. Cleavage of proBDNF by tPA/plasmin is essential for long-term hippocampal plasticity. *Science*. 306:487-491.

2) Regarding line 419: The authors do not show a change in the ratio. This would mean that more mBDNF appears on cost of proBDNF, but the authors show an increase in the ratio because they see more mBDNF. That's a difference. This does not mean that this interpretation is wrong, but, as mentioned above, there are plausible other options. Please, re-write accordingly.

3) Line 602: How many animals were excluded? What was the surgery success rate?

4) The secretome data are not easy to interpret. The authors should briefly discuss the limitation of the data set. In Fig. 3b, it is not clear me whether the volcano plot represents a mean of all samples or one representative sample?

5) line 239. This is an aspect of discussion and as a comment not easy to follow.

6) The use of the term stress is often misleading and there is a certain tendency to overinterpret the data. The authors are not clearly separating between cellular stress or stress as an adaptive in vivo response.

For instance in the abstract, line 48: These findings unravel a novel mechanistic link between stress, stress adaptation and the development of psychiatric disorders. This is not true. The authors don't show a mechanistic link between stress and the development of psychiatric disorders.

Or line 85ff: We explored a possible role for secretory autophagy as a mechanism linking GC-mediated stress to the development of psychiatric disorders.

7) Figure 7b is not covered by data.

Robert Blum

Reviewer #2 (Remarks to the Author):

In this manuscript, Martinelli and Anderzhanova et al describe the involvement of stress response in relationship to FKBP51 on secretory autophagy. Autophagy and stress response are central homeostatic regulators. Glucocorticoids are central players in stress response. Here, the authors use a number of different cell lines and manipulations to investigate the role of FKBP51 in secretory autophagy and find that FKBP51 forms complexes with some key players involved in this pathway. In addition, FKBP51 levels and/or GR activity through Dex treatment regulate secretory autophagy-related proteins. They complement these findings using microdialysis in wt and FKBP5 knockout mice exposed to stress, which revealed impaired release of CTSD, MMP9 and mBDNF into the interstitial fluid, similar to what was found following treatment with an autophagy inhibitor (ULK1i). Finally, they showed that increased FKBP51 elevates release of these same factor from microglial cells and that SAFit1 treatment reduces this. Overall, it is an interesting story, but there are major issues with the writing and data interpretation that weaken enthusiasm for the work. In addition, the overall novelty is considered moderate. Specific comments can be found below, which would strengthen the paper:

- The overall writing is not cohesive. The title does not well represent the paper, the introduction does not flow into the results, the results and discussion both have information that should be in the introduction, and the discussion has information that is more results and does not summarize the whole story well. Major revisions are needed.
- Conclusions are drawn from the co-IP data that cannot be made. Co-IPs will reveal complexes and are not quantitative. Other direct methods of binding need to be used to make statements about direct interactions between proteins. In addition, it does not appear that the lysates were precleared with beads to ensure only specific complex interactions are measured, as many proteins can stick to the beads directly.
- Do other GR-regulated proteins cause the same increase in secretory autophagy?
- Why were microglial cells selected?
- A known positive regulator of secretory autophagy should be used to compare to the results found in this paper for FKBP51 and Dex treatment
- Fig 1: Align WB strips in 1j panel
- Fig 1i: Blots look similar to S2
- Fig 1c: Appear to be the same blots are used for S1a (FKBP51, SEC22B)?
- Line 184-185: Seems to be out of place, proteins shown here look to be part of previous set of experiments (first section in results).
- Line 198: Define WT cells
- Line 214: Authors did not discuss CTSF despite it represented the biggest fold change in Fig 3c.
- Table 1: Must include references

- Fig 6: Was a control ect protein used?
- Fig S4: Describe the vehicle used in the experiment. Indicate meaning of lines (inhibitor)
- Readouts of Dex treatments need to be shown throughout. Charcoal stripped media should also be used to remove the effects of FBS hormones.
- Confirmation of FKBP51 levels and GR activity should be confirmed in each cell line being used.
- Reviews are referenced, where original manuscripts should be referenced instead. The introduction would benefit from additional references. Some references are misleading, for example Ref 1, which does not mention bipolar or schizophrenia, and Ref 8, which is not the first time secretory autophagy is described.
- Baf should be defined in the results section.
- Line 395: Secretory pathway is activated only after prolonged or excessive stress? This does not support their stress paradigm since the authors describe footshock as an acute stressor.
- Line 404-408, 412-414: missing references
- Line 428-429: It mentions NMDAR without explaining its relation to BDNF or synaptic plasticity.
- Line 448: Indicate type of stress (GC-induced, acute or both)
- Line 443: Need to discuss the "contrasting" findings- "However, despite some consistent findings, other studies report incongruent or contrasting results".
- Methods are missing for some technical aspects, including descriptions of the experiments for Figure 6 and Baf treatments.
- Authors must include exact number of n values for in vitro and in vivo experiments. Also, report the number of times the experiment was repeated, or replicates included in the final analysis.
- Mention if all animal procedures followed standard policies animal care. Include age, sex and number of animals used for each experiment. Overall, the Ns are very low for these studies and should be increased. Sex of the mice should also be considered as an independent variable. Time of day for the experiments should be carefully described in the methods
- Explain the rationale for choosing one-tailed over two-tailed unpaired t tests.
- Author Contributions: Properly indicate which type of experiments were performed by KH, TB, JH, MLP and LJ. Please indicate contribution of Felix Hausch, Christoph W Turck, Alexandra Philipsen, Mathias V. Schmidt, and Bernhard Kuster.
- All references should be in the same format. (Some include web link and DOI, see #26 and 35)
- Fig S3: Equal amount of Gapdh protein cannot be appreciated in the figure.

Reviewer #3 (Remarks to the Author):

FKBP51 (gene name: FKBP5) is a glucocorticoid (GC) receptor binding protein, which acts as a co-chaperone of heat shock protein 90 (HSP90) and regulates GC-mediated stress. This protein is also known to be associated with mental disorders.

The authors of this study showed previously that GC-mediated stress leads to the activation of macroautophagy, which is regulated by FKBP51 (Gassen et al., *pLos Med.*, 2014). In the present study, they show that GC induces another type of autophagy called secretory autophagy and that FKBP51 plays an important role in this secretory autophagy by interacting with specific SNARE proteins (Fig. 1). They further show that MM9 is a novel cargo molecule of GC-mediated secretory autophagy (Fig. 3), that FKBP51 is critical for secretion of MMP9 (Fig. 4), and that the MMP secretion plays an important role in BDNF maturation (Fig. 4, 5).

The authors demonstrated these results using FKBP51 overexpressed HEK-293 cells, Atg5 KO and FKBP-5 KO cell lines as well as FKBP5 KO mice. The story is novel, logical and supported by an impressively large amount of data which appear to be sound and quite convincing. I have two concerns however.

1. The authors provide strong cellular evidence that FKBP51 is critical for secretory autophagy of MMP9 leading to BDNF maturation. However, I feel that the biological significance for this role of FKBP51 in living mice is missing or obscure.

As the authors stated in Abstract, BDNF is essential for synaptic plasticity. Thus, I would imagine that the reduced BDNF maturation in the stress response by FKBP5 KO has strong a strong behavioral phenotype in the BDNF-related behavior, such as learning and memory. However, previous study seems to fail to show such phenotype in the FKBP5 KO mice. For instance, O'Leary et al. (2011, pLos One) described that FKBP51 KO mice showed antidepressant behavior without affecting cognition and other basic motor functions. This previous result does not seem to be consistent with the proposed function of FKBP51 in stress-induced BDNF maturation. The authors of the present study should show some sort of behavioral or neurological phenotype associated with the reduced BDNF maturation by FKBP5 KO mice. Without such evidence, the readers of Nature Communications remain puzzled about the role of FKBP5 in stress-induced synaptic plasticity and BDNF maturation.

2. Co-IP data in Fig. 1J is NOT described correctly. Fig. 1J is labeled as "GFP-IP", which I believe is correct. Based on the figure, I believed that the authors overexpressed SEC22B as GFP-fusion protein and IPed using GFP-Ab. However, the used antibody was labeled as "FLAG-Ab". This should be "GFP-Ab". The Figure legend to Fig. 1J described this experiment as "FLAG-tagged FKBP51 co-IP (FLAG-IP). I believe that the legend should be "GFP-tagged SEC22B co-IP (GFP-IP).

Reviewer #4 (Remarks to the Author):

In this manuscript by Martinelli et al., termed " Stress-primed secretory autophagy drives extracellular BDNF maturation" the authors identify the matrix metalloproteinase 9 (MMP9) as a stress-induced secreted protein involved in the cleavage of pro-brain derived neurotrophic factor (proBDNF) to its mature form (mBDNF). The authors demonstrate the involvement of the co-chaperone FK506-binding protein 51 (FKBP51) in stress-elevated secretion of MMP9 in the mouse brain, exploiting in vivo microdialysis in WT and Fkbp5 KO mice. The importance of the autophagy machinery for the stress elevated secretion is assessed in WT mice by including a ULK1 inhibitor. The authors claim that stress-induced secretion of MMP9 is through secretory autophagy, facilitated by FKBP51.

The novel finding in this manuscript is the involvement of FKBP51 in stress-elevated secretion of MMP9 in the mouse brain, resulting in maturation of BDNF. BDNF is essential for synaptic plasticity and altered BDNF signaling is associated with stress-related psychopathology. Hence, this finding is of general interest and contributes to the understanding of MMP9 secretion in the CNS. Overall, the biochemical data are well performed and the use of proteomic methods, data mining and in vivo microdialysis reflect an extensive amount of work. Nevertheless, the manuscript has some critical shortcomings that have to be addressed before publication.

The authors have previously reported involvement of FKBP51 in glucocorticoid-mediated activation of macroautophagy (Gassen et al., 2014; doi: 10.1371/journal.pmed.1001755). However, the presented data here for the involvement of secretory autophagy in stress- induced and FKBP51 dependent MMP9 secretion is not very convincing.

Firstly, the authors use cathepsin D (CTSD) and claim it is " the established secretory autophagy cargo" (page 5 line 117 and 118, page 12 line 287 and page 15 line 362). In the given reference for this claim (Kimura et al., 2017; doi: 10.15252/embj.201695081), CTSD is listed as a candidate secretory autophagy cargo in a proteomic screen. The established and most studied secretory autophagy cargo is interleukin-1beta (IL-1 β) (see e.g. New and Thomas 2019; doi: 10.1080/15548627.2019.1596479). It thus seems strange to not use IL-1 β and the authors should explain their choice. Interestingly, cathepsin B was shown to be secreted along with IL-1 β (Dupont et

al., 2011; doi: 10.1038/emboj.2011.398). The authors do show the presence of CTSD in a co-IP of ectopically expressed FLAG-TRIM16 in SH-SY5Y cells (Fig. 1e). However, that does not necessarily mean that TRIM16 binds to CTSD since IP-based interaction detection can reflect indirect binding of the proteins in question. Secondly, FKBP51 is a known co-chaperone of heat shock protein 90 (HSP90) and HSP90 has been assigned a key role in secretory autophagy of IL-1 β , mediating import of IL-1 β into the autophagosomal intermembrane space (Zhang et al., 2015; doi: 10.7554/eLife.11205). The authors do not address this point at all other than showing that mutating the HSP90 binding site of FKBP51 reduces a potential interaction between FKBP51 and galectin 8 (Fig. 2b). Finally, the authors show that FKBP51 is essential for the association of SEC22B with its Q-SNARE partners in SH-SY5Y cells (Fig. 1j). However, there are no data presented that demonstrate the importance of the formation of this SNARE complex for stress-induced MMP9 secretion in SIM-A9 (microglia) cells or mice. Detailed comments and suggestions are included below.

Major comments:

- 1) Figure 1b,c,d and e and figure S1 a and b: Interaction between FKBP51 and SEC22B or TRIM16 is implicated from reciprocal co-IPs. To demonstrate a direct binding between FKBP51 and SEC22B or TRIM16 the authors could use a GST-pulldown assay with labeled in vitro translated proteins. Furthermore, co-localization images in cells to visualize FKBP51 together with SEC22B or TRIM16 would be helpful. The same applies for the implied interaction between TRIM16 and CTSD.
- 2) For the blots in figure 1d, e and g, which form of CTSD is shown/recognized by the antibody? CTSD exists in different forms with the inactive precursor of the enzyme, procathepsin D, being cleaved, resulting in different forms of mature/active cathepsin D.
- 3) Figure 1l- "Schematic overview of the interactions of FKBP51 in the secretory autophagy pathway": Here FKBP51 is shown to interact with GAL8 but no data have yet been presented to show this. TRIM16 binds to GAL8 and the figure should indicate that. Furthermore, HSP90 is shown as a binding partner of FKBP51 but HSP90 is not included in the blots of any of the IPs in figure 1. Furthermore, there are no data presented indicating the importance of FKBP51 for transfer of the TRIM16-cargo (CTSD) to the autophagosome or data indicating that HSP90 is not present. TRIM16 association to SEC22B is independent of FKBP51 according to figure 1e. And there are no comments or experiments addressing how CTSD, that normally resides in lysosomes, is translocated into the lumen of autophagosomes prior to its secretion. Therefore, the claim on page 6 line 148-149. "From these data, FKBP51 results to be involved in several key steps of the secretory autophagy pathway (Fig 1l)", appears as an overstatement.
- 4) In figure 2c and 2d the authors use a tandem tagged (mRFP-GFP) galectin 3 (tfGal3) in SH-SY5Y cells to monitor lysosomal damage. The reduction of the GFP signal is a result of acidification of tfGal3. Gal3 is recruited to damaged lysosomes and Gal3 becomes acidified through lysophagy. Lysophagy involves autophagosomal engulfment of damaged lysosomes that subsequently become degraded by fusion with intact lysosomes (Maejima et al., 2013; doi: 10.1038/emboj.2013.171). LLOMe induces lysosomal damage that culminates in lysophagy and Dex appears to be able to do the same. Inhibition of lysosomal acidification with BafA1 abolishes the effect. Therefore these data actually show degradation of Gal3 on damaged lysosomes through autophagy. The authors should comment on how they envision the effect of dex on lysosomes and how this relates to secretory autophagy.
- 5) In figure 2 e) and 2 f) SIM-A9 secretion of CTSD in response to LLOMe and dex treatment, respectively should also include BafA1 treatment to determine if the secretion is dependent on functional lysosomes or not.
- 6) In figure 4 the dex induced MMP9 secretion should be shown in FKBP51 KO SIM-A9 cells as well to complement the in vivo results in figure 5. Furthermore, in order to link MMP9 secretion to secretory autophagy the authors could use siRNA knockdown of TRIM16 or SEC22B in these cells. The presence of MMP9 in an IP of TRIM16 or co-localization study of MMP9 with TRIM16 in cells would also be desirable.
- 7) Figure 7-"Schematic representation of the findings and proposed model": In a, TRIM16 is shown to interact with MMP9. Again, there are no data in the manuscript that demonstrate this interaction and this schematic drawing is thus not accurate.

Minor comments:

- 1) In Supplementary Table S1 of 29 identified interactors of FKBP51 in HEK293 cells, HSP90 is not listed. The authors should comment on that.
- 2) Page 5 line 114-115: "...the interaction of FKBP51 with SEC22B (Fig. 1b,c), previously only deduced via differential centrifugation¹³". Please clarify, since the cited reference (Kimura et al., 2017; doi: 10.15252/embj.201695081) appears not to have any data implying interaction between FKBP51 and SEC22B.
- 3) In figure 1g, FKBP51 levels are not increased with 100 nm Dex. This is in contrast to supplementary figure 1d showing quantified increase of FKBP51 expression with 100 nm Dex. The authors should comment on this and include an image of a representative blot for the figure S1d.
- 4) In figure 2a the authors should include a representative image of the blot.
- 5) The figure legends should indicate which cell type is used. This is lacking in several of the figure legends and should be clearly indicated since both HEK293, SH-SY5Y and SIM-A9 cells are used.
- 6) Methods-ELISA page 22. In the description, ELISA kit product numbers are indicated and not antibody product numbers as the text indicates. This should be corrected. Furthermore, in line 557 it says "Amounts of Il1b were detected with a plate reader". Il1b is not included in this study.
- 7) In the figure legend to figure 2, page 40 line 028, dexamethasone should be written after 300 nM.
- 8) In figure legend to figure 5 page 41, there is a typo error in line 056. ILK1 inhibitor, should be ULK1 inhibitor.
- 9) The discussion section is rather long and should be more focused on the obtained data in this study.

Reviewer #5 (Remarks to the Author):

The manuscript by Martinelli et al. "Stress-primed secretory autophagy drives extracellular BDNF maturation" described the mechanism of enhancement of secretory autophagy by glucocorticoid-induced stress. The authors use interactomics and secretome analysis by mass spectrometry to identify proteins involved in the process and propose an elegant step-by-step mechanistic model validated by several other methods. The paper is well written, the findings are novel and this reviewer supports the publication in Nature Communications which will allow these results to reach a broad readership. There are a few minor concerns, mostly technical in nature, that should be addressed before acceptance.

Detailed comments:

1. Since TRIM16 was confirmed to be an interactor of FKBP51 by Western blot, but not originally found in the MS dataset, can the MS data be researched and perhaps TRIM16 peptides can be found (maybe with the help of the inclusion list?)
2. For the interactome analysis, the transfection with a vector containing FLAG only was used. Can the authors include more description of how the control (unspecific) binders were eliminated? The only explanation I could find is in lines 651-651, that the proteins overlapping from all four replicates were counted as interactors. This is not sufficient.
3. For the secretome analysis, the media supplemented with FBS was used throughout the whole experiment. Is that correct? If so, how was the signal suppression by the overwhelming amount of protein handled? Was the albumin removed? This was not an issue for detection because the labeling was used, but signal suppression would be an issue anyway. Can the authors comment on that?
4. Another useful clarification of the secretome analysis would be the comparison of growth rates and cell death between the wild type and the Atg5 KO cells. Are they identical in this respect, and if not, how was the data normalized?

Point by point rebuttal to the reviewers' comments

Reviewer #1

We thank Robert Blum (Reviewer #1) for his constructive and insightful comments. We addressed his valid suggestions by performing additional experiments and tried to clarify imprecisions with additional information.

Reviewer #1 (Remarks to the Author):

The paper by Martinelli Anderzhanova et al presents new insights in regulation of secretory autophagy, identifies new and important regulatory proteins of secretory autophagy. The study shows for the first time, how one important cargo protein of secretory autophagy, the matrix metalloproteinase 9 (MMP9) can contribute to more behavior-related, extracellular abundance of mature BDNF.

BDNF is one of the most important key proteins in synaptic plasticity, it can be stored in synapses and undergoes activity-dependent secretion. BDNF is involved in diverse synaptic processes, including synapse maturation, synapse refinement, synaptic transmission and even pre- and postsynaptic LTP. Two BDNF isoforms are known to regulate synaptic transmission, the mature BDNF, a homodimeric protein with high affinity to TrkB, and proBDNF, an isoform carrying the so called pro-domain. Even the cleaved pro-domain has been shown to be involved in synaptic processes. proBDNF shows high-affinity to p75, another neurotrophin receptor. There is an ongoing debate about proBDNF secretion and processing.

In this important study, the authors show that FKBP51, a stress responsive co-chaperone is critically involved in secretory autophagy. Notably, MMP9 is a cargo of these granules and undergoes regulated secretion, possibly for the local cleavage of proBDNF to mBDNF. The finding that cellular stress-related release from autophagosome-like granules/vesicles is associated with extracellular mBDNF abundance is credible.

The design of the study is straightforward and conceptually strong. The experiments are convincing and there is a lot of important, new and relevant information available. For instance, the interactome of FKBP51 is well worked out and the experiments help better understand how FKBP51-positive autophagosomes behave within the trafficking pathway. Much of the work has been done in cell lines (SH5YSY, neuroblastoma-like cell line; Hek293, SIM-A9, microglia-like), but the detailed information about key proteins in the secretory autophagosome (SEC22B, RACK1, UBC12) pathway will help to find out how secretory autophagy is acting in neurons, at synapses or between cell types (microglia) at synapses.

The in vivo experiments clearly show, with a new approach, direct determination of behavior-related secretion of BDNF isoforms. However, we do not learn from which

cells BDNF is secreted, an aspect beyond the scope of this study. Nevertheless, the data convincingly reveal that MMP9 and BDNF appear in the prefrontal cortex in the course of a behavioral test and the data show that MMP9 contributes to more mBDNF.

I'm sure that these proof-of-principle experiments will motivate many researchers in the field to look again at the fundamental biology of synaptic BDNF in 'real' learning and memory paradigms.

The methodology is of highest quality. I think that this paper is of substantial importance. For me, there are some major concerns that have to be addressed before it can be considered for publication in Nature communication.

Major comments:

- (1) The authors write (line 88): ...an increase in cleavage of pro-brain-derived neurotrophic factor (proBDNF) to its mature form (mBDNF) both in vitro and in vivo.

However, the data do not show that proBDNF cleavage leads to less proBDNF and more mBDNF. The data show the same amount of proBDNF (in vitro-ELISA) and in vivo (Immunoblotting; but size is not given). If there is cleavage on cost of proBDNF, Western analysis should show less proBDNF at its full size relative molecular weight and more mBDNF at 13 kDa. The interpretation and discussion of the data should be in line with the data.

Response:

Dexamethasone treatment leads to an increase in BDNF expression, as shown in the supplementary fig. S5, and to a consequential increased proBDNF secretion, resulting from ELISA (Fig.4b) and microdialysate data (Fig. 5e and I and supported by the following table showing the additional statistics of the 2-way ANOVA analysis performed in the microdialysates experiment where proBDNF's expression is dependent on the time factor but independent from the genotype or treatment factor). This enhanced expression could explain the almost unchanged levels of proBDNF, as it is more expressed and secreted while it is degraded extracellularly.

2-way ANOVA analyses of microdialysates

	Ctsd in WT/KO	MMP9 in WT/KO	proBDNF in WT/KO	mBDNF in WT/KO
Time x genotype	<0.0001	<0.0001	0,2636	0,0062
Time	<0.0001	<0.0001	<0.0001	0,0851
Genotype	0,0002	<0.0001	0,5137	0,0294
Subject	0,6859	0,9580	0,7471	0,2313
	Ctsd in ctr/AI	MMP9 in ctr/AI	proBDNF in ctr/AI	mBDNF in ctr/AI
Time x genotype	0,0095	<0.0001	0,8101	0,0152
Time	<0.0001	<0.0001	0,0011	<0.0001
Genotype	0,0346	0,0002	0,2327	0,0125
Subject	0,7641	0,3262	0,3641	0,4353

Regarding the molecular weight of mBDNF, we observe proBDNF and the physiologically active mBDNF dimer in the microdialysates as 32-kDa- and 26-kDa- signals, respectively. We do identify a signal at 13 kDa corresponding to monomers of mBDNF. However, this signal is below the LOD and for this reason not quantifiable (see figure below showing a representative SimpleWestern blot quantified for experiments shown in Fig. 6: secreted proBDNF or BDNF from DMSO-treated mouse organotypic brain slices.)

- (2) Fig. 3: Secretome. proBDNF and mBDNF should appear in the secretome, but the authors do not show the data and do not discuss it (or I have overseen it). My question is: Is there more or less pro-domain or mBDNF in the secretome of SIM-A9 cells after Atg5KO or is there an explanation why there is not much BDNF in the secretome? In line with the overall concept of the study, one assumption would be that if BDNF is in a different vesicle and is not co-released with MMP-9, one would expect to see

more pro-domain in case of Dex/Atg5KO. Is this really extracellular cleavage on cost of proBDNF or are there other explanations?

Response:

ProBDNF and mBDNF indeed have not been detected in the secretomics experiment. A general drawback of untargeted, discovery-driven MS analysis applied here is the fact that the selection of full peptides on MS1 level for the fragmentation and subsequent identification on MS2 level is intensity-based. This means that only the x most intense peptide ions in a MS1 spectrum are selected for further processing in MS2 (where x might range from 5 to 20 depending on the MS method used). Consequently, low intense peptides (derived from proteins of low intensity) might be missed from mass spectrometric detection although they have been present in the sample. This has most certainly happened to proBDNF and mBDNF, which, however, have been detected in a targeted assay (fig. 4 and supplementary fig. S4c). Furthermore, the fact that mBDNF levels are significantly reduced upon treatment with MMP9i (which inhibits MMP9's enzymatic activity only in the extracellular space), demonstrates that the proBDNF to mBDNF cleavage occurs extracellularly and at the hand of MMP9.

- (3) Fig 4: Important data depend on SIM-A9 cells, a microglia-like cell line. The data suggest that the cells produce BDNF and secrete it. However, western analysis is missing. Western blotting data would help to find out whether proBDNF is intracellular and then secreted as proBDNF. It can well be that this is secretion or abundance of cleaved pro-domain versus mBDNF. In Western blotting after SDS-PAGE, mBDNF should appear at 13 kDa, the cleaved pro-domain should run close to 15 – maybe 20 kDa, the uncleaved pro-domain may be expected, in its glycosylated form at about 32-34 kDa.

Immunoblotting from supernatants, maybe after IP or after protein concentration, should answer this important aspect. It may be that the authors find other or new anti-pro/anti-BDNF immunoreactive bands. This experiment can help to better understand the overall data set.

Response:

Regarding the size of proBDNF and mBDNF please refer to our answer to comment 1.

For the purpose of our investigation, we found ELISA assays more sensitive and reliable and therefore favored this method. We used in some cases Western blot analyses (see supplementary Fig. S4 f and g) which confirmed our findings with the additional information of the protein sizes. However, the technical steps necessary to remove the high abundance of serum proteins contained in the culture medium would interfere with a direct comparison between the intracellular and extracellular protein quantifications. Furthermore, the extracellular quantification focuses on exactly our point of interest of whether MMP9 leads to extracellular cleavage of BDNF, since we have two observation time points: before and after treatment. While the detection of

further or novel variants of BDNF represents an interesting challenge, it might deviate from the scope of this manuscript.

- (4) Fig. 5: The authors quantified BDNF and proBDNF in microdialysates. The method is rather new and of outstanding importance. The method section is not clear here. Antibodies or other reagents are not given. The method is immunoblotting-dependent (line 286). Original data (proBDNF, BDNF, MMP9) should be shown. The authors should cite or point to another study showing how this new approach was developed and verified.

Response:

We updated the Method section in particular in reference to the recently published paper (Anderzhanova et al., 2020), where a comprehensive validation of the measurements using multiple methods was provided.

Furthermore, SimpleWestern blotting is an automated system where capillary-based immunoblotting signals are automatically quantified and normalized by the system. The shown quantifications are, therefore, the original data (outputs) of the assays. Additionally, all original raw data are made available to the editors in the final submission procedure. Regarding the information relative to the Antibodies, they are the same used for PAGE Western blotting.

- (5) In Fig. 6, after the in vivo data, the authors present the SAFit1 inhibition of FKBP51 in cell culture. The data fit better to Figure 3. Again, as mentioned above, the data are not supported by Western blotting. Here, it is ELISA detection of proBDNF or mBDNF from supernatants. ProBDNF is unchanged, but there is less mBDNF after SAFit1 inhibition. It can well be that MMP9 is acting at a different place. Or another yet unknown MMP9-associated process leads to more secretion of mBDNF. This might explain that there is almost no change in proBDNF in Fig. 4, 5e and 5i. Fig. 6. Anyhow, in Fig. 5, it is very clear that mBDNF goes up, in vivo, extracellular, regulated and behavior-related. This direct verification of regulated, in vivo BDNF secretion is a very important finding.

Response:

Please refer to our previous responses to comments 1 and 3. The main proof of the extracellular cleavage is the fact that by inhibiting the extracellular enzymatic activity of MMP9 via MMP9i, the mBDNF levels do not increase (Fig. 4f). We do not have any evidence to suppose a novel function of MMP9 that acts on mBDNF secretion, since its extracellular enzymatic activity is well characterized and there is no structural feature of MMP9 that leads to hypothesize such a function.

Regarding Fig. 6, we followed reviewer #1's suggestion and added the FKBP51 overexpression and SAFit1 treatment data to the dataset of Fig. 4, which we think reviewer #1 was referring to and we think would fit best.

Minor

- (6) The idea of functional extracellular cleavage of proBDNF is quite old. Please cite the corresponding landmark paper: Pang, P.T., H.K. Teng, E. Zaitsev, N.T. Woo, K. Sakata, S. Zhen, K.K. Teng, W.H. Yung, B.L. Hempstead, and B. Lu. 2004. Cleavage of proBDNF by tPA/plasmin is essential for long-term hippocampal plasticity. *Science*. 306:487-491.

Response:

We tried to add this information, but it felt really forced as we address the BDNF cleavage from an unbiased finding of MMP9 secretion. We never exclude the existence of other proteolytic enzymes, nor we claim to be the first to describe BDNF cleavage by MMP9. On the contrary, we focus on the secretion of MMP9, thus the cleavage of BDNF by other enzymes (whether plasmin or other metalloproteases) is out of our focus.

- (7) Regarding line 419: The authors do not show a change in the ratio. This would mean that more mBDNF appears on cost of proBDNF, but the authors show an increase in the ratio because they see more mBDNF. That's a difference. This does not mean that this interpretation is wrong, but, as mentioned above, there are plausible other options. Please, re-write accordingly.

Response:

As stated in the responses to comments 1, 3 and 5, even though we acknowledge the reviewer's concern regarding the lack of decrease in extracellular proBDNF, there is indeed proof that the increase in extracellular mBDNF occurs on cost of proBDNF since this process can be impaired via inhibition of MMP9, that is responsible for such conversion. The increase in mBDNF/ proBDNF ration remains therefore valid, as only mBDNF increases and not proBDNF, which is the important aspect here given the opposing roles of the two isoforms in synaptic plasticity.

- (8) Line 602: How many animals were excluded? What was the surgery success rate?

Response:

We started our experiments with six mice per experimental group, taking into account a possible loss of 1-2 mice per group. Despite having a surgery success rate of 100% we had to exclude some animals during perfusion. Sometimes microdialysis probes stop working, which is a strong indication to exclude the animals. Most of the microdialysate data derive from four mice per group and from five mice for some groups (individuals indicated by dots in graphs).

- (9) The secretome data are not easy to interpret. The authors should briefly discuss the limitation of the data set. In Fig. 3b, it is not clear me whether the volcano plot

represents a mean of all samples or one representative sample?

Response:

The major challenge of secretome analyses is the large abundance of serum proteins in the medium that hide the signal of secreted proteins. To avoid this problem, we used an innovative technique developed by Eichelbaum and colleagues (Eichelbaum et al., 2014) where proteins were labelled with AHA (L-Azidohomoalanin), an amino acid analog of methionine, then purified via click-it reaction and subsequently analyzed via LC-MS/MS. The AHA labelling procedure presents some limitations such as the fact that proteins low in methionine might have been missed from initial enrichment by click chemistry. Similarly, methionine-containing peptides are missing from the dataset. In such way, the chance to detect e.g. small proteins is lower. The coverage of proteins (based on peptides) is lower which can makes quantification and sometimes identification somehow difficult.

Finally, a general MS drawback is that a data-dependent acquisition is used here. This means that only a sampling of the most intense peptides for further fragmentation and identification has happened. Very low intense secreted proteins might have been missed from detection. Overall, however, this method allowed us to have a more accurate sampling of the secretome compared to the traditional, non-labelled analyses.

Regarding the volcano plot data, each dot represents the sample mean. This information was added to the legend.

(10) line 239. This is an aspect of discussion and as a comment not easy to follow.

Response:

We removed that last sentence and integrated it into the discussion.

(11) The use of the term stress is often misleading and there is a certain tendency to over-interpret the data. The authors are not clearly separating between cellular stress or stress as an adaptive in vivo response.

For instance in the abstract, line 48: These findings unravel a novel mechanistic link between stress, stress adaptation and the development of psychiatric disorders. This is not true. The authors don't show a mechanistic link between stress and the development of psychiatric disorders.

Or line 85ff: We explored a possible role for secretory autophagy as a mechanism linking GC-mediated stress to the development of psychiatric disorders.

Response:

We acknowledged the overstatement and edited the whole manuscript with this in mind. We better differentiated between evidence-based findings and speculations. We also tried to better define stress, as it is a very broad and variable concept and topic that has different interpretations in the different scientific fields.

(12) Figure 7b is not covered by data.

Response:

That is correct. Figure 7b is a proposed model of the autophagic stress-response dynamics. We took the possible misunderstanding into consideration and added this information to the legend text for clarity.

Reviewer #2

We thank Reviewer #2 for their detailed feedback. We edited the manuscript according to their suggestions in a way that we hope will result more cohesive to the reader. We also addressed technical aspects with additional data and control experiments.

Reviewer #2 (Remarks to the Author):

In this manuscript, Martinelli and Anderzhanova et al describe the involvement of stress response in relationship to FKBP51 on secretory autophagy. Autophagy and stress response are central homeostatic regulators. Glucocorticoids are central players in stress response. Here, the authors use a number of different cell lines and manipulations to investigate the role of FKBP51 in secretory autophagy and find that FKBP51 forms complexes with some key players involved in this pathway. In addition, FKBP51 levels and/or GR activity through Dex treatment regulate secretory autophagy-related proteins. They complement these findings using microdialysis in wt and FKBP5 knockout mice exposed to stress, which revealed impaired release of CTSD, MMP9 and mBDNF into the interstitial fluid, similar to what was found following treatment with an autophagy inhibitor (ULK1i). Finally, they showed that increased FKBP51 elevates release of these same factor from microglial cells and that SAFit1

treatment reduces this. Overall, it is an interesting story, but there are major issues with the writing and data interpretation that weaken enthusiasm for the work. In addition, the overall novelty is considered moderate. Specific comments can be found below, which would strengthen the paper:

(1) The overall writing is not cohesive. The title does not well represent the paper, the introduction does not flow into the results, the results and discussion both have

information that should be in the introduction, and the discussion has information that is more results and does not summarize the whole story well. Major revisions are needed.

Response:

We thoroughly revised the paper with this comments and the editor's feedback in mind and hope that reviewer #2 will find the new version more coherent and readable.

- (2) Conclusions are drawn from the co-IP data that cannot be made. Co-IPs will reveal complexes and are not quantitative. Other direct methods of binding need to be used to make statements about direct interactions between proteins. In addition, it does not appear that the lysates were precleared with beads to ensure only specific complex interactions are measured, as many proteins can stick to the beads directly.

Response:

We acknowledge that co-IP data do not necessarily implicate a direct interaction, therefore we replaced the term "interaction" with "association", meaning the formation of a complex via direct or indirect interaction. In fact, an indirect interaction would not contradict our model and our focus remains to show that FKBP51 is necessary for the complex formation with SEC22B and the SNARE proteins that leads to the secretion of the secretory autophagy cargo. However, to address Reviewer #2's demand, we also performed pull-down experiments that demonstrate that FKBP51 can directly interact with TRIM-16 in vitro (Supplementary Fig. S1c).

Regarding the technical aspects of the IP's specificity, we precleared the beads with BSA (bovine serum albumin) to omit unspecific binding of proteins. This way of proceeding allows an increased specificity for the target protein (i.e. for the antibody), while reducing unspecific binding to the bead material. In addition, to enhance specificity we performed a selective elution of IP-material from antibody-loaded beads using peptide competition with excessive amounts of FLAG peptide.

- (3) Do other GR-regulated proteins cause the same increase in secretory autophagy?

Response:

We have not tested other GR-regulated proteins, since we selected FKBP51 based on the newly found interaction with SEC22B. Although it would certainly be interesting to assess whether stress can affect the secretory pathway via other proteins, the fact that FKBP51's absence (in SH-SY5Y KO) or inhibition (via SAFit1) suffices to impair this pathway, suggests that FKBP51 has a rather unique and specific role that we think is hardly a general feature of other GR-regulated proteins. A possible examination of the role of other GR-regulated proteins in the secretory pathway is a laborious project (or projects) and is out of the scope of this manuscript.

- (4) Why were microglial cells selected?

Response:

Our interest focuses on stress-related psychiatric disorders, therefore we wanted to analyze this mechanism in brain cells, and to ensure a proper readout for our experiments we selected microglia cells as they are the main secretory cells in the brain.

- (5) A known positive regulator of secretory autophagy should be used to compare to the results found in this paper for FKBP51 and Dex treatment

Response:

We used L-leucyl-L-leucine methyl ester (LLOMe) as a positive regulator of secretory autophagy (Fig. 2 c and e) since it is the best characterized secretory autophagy inductor (Kimura et al., 2016). We added this information to the manuscript for clarity (line 199-201).

- (6) Fig 1: Align WB strips in 1j panel

Response:

They are now aligned.

- (7) Fig 1i: Blots look similar to S2

Response:

They look indeed similar as they represent the same proteins and same conditions but in two different cell lines. However, a close look reveals several small differences that confirm that they are indeed two different blots. (little cut in the FKBP51 and SNAP29 bands of Fig. 1i, shadows of bands in the FLAG conditions of STX3 and SEC22B in fig. S2, different shape of all the other bands).

- (8) Fig 1c: Appear to be the same blots are used for S1a (FKBP51, SEC22B)?

Response:

FKBP51 and SEC22B are indeed the same. We repeated them in the supplements for completion. We realized it can lead to confusion and therefore proceeded to remove SEC22B from the supplementary figure (S1a), but kept FKBP51 since it is the immunoprecipitated protein.

- (9) Line 184-185: Seems to be out of place, proteins shown here look to be part of previous set of experiments (first section in results).

Response:

We did not use SIM-A9 cells for the first section of the results. However, we acknowledge that this sentence is not well incorporated in the paragraph and, therefore, rewrote it.

(10) Line 198: Define WT cells

Response:

WT SIM-A9 cells underwent the same transfection procedure as the Atg5 KO cells but without the gRNA targeting Atg5. SEC22B and FKBP5 KO SIM-A9 cells were generated with the Alt-R CRISPR-Cas9 system from Integrated DNA Technologies. WT control cells were identified by WB after single-cell cloning procedures and therefore underwent the same transfection and isolation procedure as the KO cells.

(11) Line 214: Authors did not discuss CTSF despite it represented the biggest fold change in Fig 3c.

Response:

The deeper investigation of the effect of stress on the secretion of cathepsins is out of the scope of our manuscript, but is part of a follow up study (Niemeyer et al., in preparation).

(12) Table 1: Must include references

Response:

Due to the limit of allowed references in the main manuscript, the references of Table 1 can be found as Supplementary references file. We added the missing link to it in the Table legend (lines 1308-1309). We ask the Editor for suggestions for the best format for this purpose.

(13) Fig 6: Was a control ect protein used?

Response:

As control, an empty vector was used (same vector as ect. FKBP51).

(14) Fig S4: Describe the vehicle used in the experiment. Indicate meaning of lines (inhibitor)

Response:

We added the information regarding the used vehicle to the method "Treatments" section (line 545). We also indicated the meaning of lines in panel c.

(15) Readouts of Dex treatments need to be shown throughout. Charcoal stripped media should also be used to remove the effects of FBS hormones.

Response:

In our experience, glucocorticoids contained in FBS are in too low concentrations to elicit any GR activation. However, to ensure that this is true for the mechanism analyzed in this manuscript, we performed additional experiments with charcoal stripped medium. We measured secreted CTSD and MMP9 via ELISA. The figure below shows the results from WT and Atg5 KO SIM-A9 cells treated with vehicle or 300 nM of dexamethasone for four hours and cultured in FBS- or charcoal stripped serum (CSS)-supplemented culture medium for 24 h hours (n=3).

From these results, we can observe a rather diminished response to GR activation in the presence of complete FBS and, as a consequence, the significant results obtained using complete FBS do rather strengthen our output and conclusions.

In the tables below, the complete statistics regarding the multiple comparison are shown, confirming that complete FBS does not affect the outcome of our experiments.

2-way ANOVA multiple comparisons

CTSD					
Tukey's multiple comparisons test	Mean Diff.	95.00% CI of diff.	Below threshold?	Summary	Adjusted P Value
WT vehicle:FBS vs. WT vehicle:CSS	0,1580	-0.9511 to 1.267	No	ns	0,9995
WT vehicle:FBS vs. WT Dex:FBS	-1,245	-2.354 to -0.1354	Yes	*	0,0223
WT vehicle:FBS vs. WT Dex:CSS	-2,036	-3.145 to -0.9273	Yes	***	0,0002
WT vehicle:FBS vs. Atg5 KO vehicle:FBS	0,2547	-0.8543 to 1.364	No	ns	0,9910
WT vehicle:FBS vs. Atg5 KO vehicle:CSS	0,2127	-0.8964 to 1.322	No	ns	0,9969
WT vehicle:FBS vs. Atg5 KO Dex:FBS	0,01396	-1.095 to 1.123	No	ns	>0.9999
WT vehicle:FBS vs. Atg5 KO Dex:CSS	0,1448	-0.9643 to 1.254	No	ns	0,9997
WT vehicle:CSS vs. WT Dex:FBS	-1,403	-2.512 to -0.2934	Yes	**	0,0086
WT vehicle:CSS vs. WT Dex:CSS	-2,194	-3.303 to -1.085	Yes	****	<0.0001
WT vehicle:CSS vs. Atg5 KO vehicle:FBS	0,09677	-1.012 to 1.206	No	ns	>0.9999
WT vehicle:CSS vs. Atg5 KO vehicle:CSS	0,05472	-1.054 to 1.164	No	ns	>0.9999
WT vehicle:CSS vs. Atg5 KO Dex:FBS	-0,1440	-1.253 to 0.9651	No	ns	0,9997
WT vehicle:CSS vs. Atg5 KO Dex:CSS	-0,01315	-1.122 to 1.096	No	ns	>0.9999
WT Dex:FBS vs. WT Dex:CSS	-0,7918	-1.901 to 0.3172	No	ns	0,2736
WT Dex:FBS vs. Atg5 KO vehicle:FBS	1,499	0.3902 to 2.608	Yes	**	0,0048
WT Dex:FBS vs. Atg5 KO vehicle:CSS	1,457	0.3481 to 2.566	Yes	**	0,0061
WT Dex:FBS vs. Atg5 KO Dex:FBS	1,258	0.1494 to 2.368	Yes	*	0,0205
WT Dex:FBS vs. Atg5 KO Dex:CSS	1,389	0.2803 to 2.498	Yes	**	0,0093
WT Dex:CSS vs. Atg5 KO vehicle:FBS	2,291	1.182 to 3.400	Yes	****	<0.0001
WT Dex:CSS vs. Atg5 KO vehicle:CSS	2,249	1.140 to 3.358	Yes	****	<0.0001
WT Dex:CSS vs. Atg5 KO Dex:FBS	2,050	0.9412 to 3.159	Yes	***	0,0002
WT Dex:CSS vs. Atg5 KO Dex:CSS	2,181	1.072 to 3.290	Yes	****	<0.0001
Atg5 KO vehicle:FBS vs. Atg5 KO vehicle:CSS	-0,04205	-1.151 to 1.067	No	ns	>0.9999
Atg5 KO vehicle:FBS vs. Atg5 KO Dex:FBS	-0,2408	-1.350 to 0.8683	No	ns	0,9935
Atg5 KO vehicle:FBS vs. Atg5 KO Dex:CSS	-0,1099	-1.219 to 0.9992	No	ns	>0.9999
Atg5 KO vehicle:CSS vs. Atg5 KO Dex:FBS	-0,1987	-1.308 to 0.9104	No	ns	0,9980
Atg5 KO vehicle:CSS vs. Atg5 KO Dex:CSS	-0,06787	-1.177 to 1.041	No	ns	>0.9999
Atg5 KO Dex:FBS vs. Atg5 KO Dex:CSS	0,1309	-0.9782 to 1.240	No	ns	0,9999

2-way ANOVA multiple comparisons

MMP9					
Tukey's multiple comparisons test	Mean Diff.	95.00% CI of diff.	Below threshold?	Summary	Adjusted P Value
WT vehicle:FBS vs. WT vehicle:CSS	0,1014	-0.4212 to 0.6241	No	ns	0,9967
WT vehicle:FBS vs. WT Dex:FBS	-0,7198	-1.243 to -0.1972	Yes	**	0,0040
WT vehicle:FBS vs. WT Dex:CSS	-1,240	-1.763 to -0.7174	Yes	****	<0.0001
WT vehicle:FBS vs. Atg5 KO vehicle:FBS	-0,08791	-0.6106 to 0.4348	No	ns	0,9987
WT vehicle:FBS vs. Atg5 KO vehicle:CSS	0,07761	-0.4451 to 0.6003	No	ns	0,9994
WT vehicle:FBS vs. Atg5 KO Dex:FBS	0,1422	-0.3804 to 0.6649	No	ns	0,9766
WT vehicle:FBS vs. Atg5 KO Dex:CSS	0,03435	-0.4883 to 0.5570	No	ns	>0.9999
WT vehicle:CSS vs. WT Dex:FBS	-0,8213	-1.344 to -0.2986	Yes	**	0,0011
WT vehicle:CSS vs. WT Dex:CSS	-1,341	-1.864 to -0.8188	Yes	****	<0.0001
WT vehicle:CSS vs. Atg5 KO vehicle:FBS	-0,1893	-0.7120 to 0.3333	No	ns	0,9028
WT vehicle:CSS vs. Atg5 KO vehicle:CSS	-0,02382	-0.5465 to 0.4988	No	ns	>0.9999
WT vehicle:CSS vs. Atg5 KO Dex:FBS	0,04080	-0.4819 to 0.5635	No	ns	>0.9999
WT vehicle:CSS vs. Atg5 KO Dex:CSS	-0,06708	-0.5898 to 0.4556	No	ns	0,9998
WT Dex:FBS vs. WT Dex:CSS	-0,5202	-1.043 to 0.002461	No	ns	0,0515
WT Dex:FBS vs. Atg5 KO vehicle:FBS	0,6319	0.1093 to 1.155	Yes	*	0,0125
WT Dex:FBS vs. Atg5 KO vehicle:CSS	0,7974	0.2748 to 1.320	Yes	**	0,0015
WT Dex:FBS vs. Atg5 KO Dex:FBS	0,8621	0.3394 to 1.385	Yes	***	0,0007
WT Dex:FBS vs. Atg5 KO Dex:CSS	0,7542	0.2315 to 1.277	Yes	**	0,0026
WT Dex:CSS vs. Atg5 KO vehicle:FBS	1,152	0.6295 to 1.675	Yes	****	<0.0001
WT Dex:CSS vs. Atg5 KO vehicle:CSS	1,318	0.7950 to 1.840	Yes	****	<0.0001
WT Dex:CSS vs. Atg5 KO Dex:FBS	1,382	0.8596 to 1.905	Yes	****	<0.0001
WT Dex:CSS vs. Atg5 KO Dex:CSS	1,274	0.7517 to 1.797	Yes	****	<0.0001
Atg5 KO vehicle:FBS vs. Atg5 KO vehicle:CSS	0,1655	-0.3572 to 0.6882	No	ns	0,9485
Atg5 KO vehicle:FBS vs. Atg5 KO Dex:FBS	0,2301	-0.2925 to 0.7528	No	ns	0,7845
Atg5 KO vehicle:FBS vs. Atg5 KO Dex:CSS	0,1223	-0.4004 to 0.6449	No	ns	0,9900
Atg5 KO vehicle:CSS vs. Atg5 KO Dex:FBS	0,06462	-0.4580 to 0.5873	No	ns	0,9998
Atg5 KO vehicle:CSS vs. Atg5 KO Dex:CSS	-0,04326	-0.5659 to 0.4794	No	ns	>0.9999
Atg5 KO Dex:FBS vs. Atg5 KO Dex:CSS	-0,1079	-0.6306 to 0.4148	No	ns	0,9952

(16) Confirmation of FKBP51 levels and GR activity should be confirmed in each cell line being used.

Response:

Dexamethasone stimulations in SIM-A9 was performed and FKBP51 levels were analyzed via western blot. Dexamethasone treatments led to a significant increase of FKBP51 levels confirming the dexamethasone-induced GR activation and increased expression of FKBP51 in this cell line. These data are shown in Supplementary Fig. S2a.

(17) Reviews are referenced, where original manuscripts should be referenced instead. The introduction would benefit from additional references. Some references are misleading, for example Ref 1, which does not mention bipolar or schizophrenia, and Ref 8, which is not the first time secretory autophagy is described.

Response:

Due to the limit in number of references that are allowed we sometimes preferred to reference the reviews to convey more information especially in reference of a broader topic such as secretory autophagy, for which there is no real first publication, but rather a first publication in which the term was coined (Jiang *et al.*, 2013). However, we now added more direct references in addition to the reviews in order to better specify our sources.

(18) Baf should be defined in the results section.

Response:

It is defined in lines 198-202.

(19) Line 395: Secretory pathway is activated only after prolonged or excessive stress? This does not support their stress paradigm since the authors describe footshock as an acute stressor.

Response:

Footshock is an acute but very strong stress. It, therefore, falls into the category excessive stress.

(20) Line 404-408, 412-414: missing references

Response:

References were added.

(21) Line 428-429: It mentions NMDAR without explaining its relation to BDNF or synaptic plasticity.

Response:

The discussion was thoroughly revised and adapted to the new data. In this optics we also incorporated the discussion about NMDAR, BDNF and synaptic plasticity in a more cohesive way.

(22) Line 448: Indicate type of stress (GC-induced, acute or both)

Response:

We better defined this point both here (line 523). However, we would like to specify that throughout the whole manuscript, when we talk about stress, we always refer to GC-mediated stress, as stated in the introduction (lines 96-97).

(23) Line 443: Need to discuss the “contrasting” findings- “However, despite some consistent findings, other studies report incongruent or contrasting results”.

Response:

The discussion was thoroughly revised and adapted to the new data. In this process, this sentence was eliminated.

(24) Methods are missing for some technical aspects, including descriptions of the experiments for Figure 6 and Baf treatments.

Response:

Methods regarding Fig. 6 can be found in the Method section “Transfections” (lines 551-558) and “ELISA” (lines 650-658).

SAFit1 and Baf treatments were indeed missing and were added in the section “Treatments” (lines 545-550)

(25) Authors must include exact number of n values for in vitro and in vivo experiments. Also, report the number of times the experiment was repeated, or replicates included in the final analysis.

Response:

The exact n values can be found in each figure legend and all information will be reported in detail in the source data table upon acceptance. However, all experiments were performed in at least three technical replicates and at least three biological replicates.

(26) Mention if all animal procedures followed standard policies animal care. Include age, sex and number of animals used for each experiment. Overall, the Ns are very low for these studies and should be increased. Sex of the mice should also be considered as an independent variable. Time of day for the experiments should be carefully described in the methods

Response:

We added the first information in the Methods section “Animal housing conditions” (lines 666-673). We specify here that all procedures were done in accordance with

European Communities Council Directive 2010/63/EU and approved by Government of Upper Bavaria.

All mice used in *in vivo* experiments were males. FKBP51-KOs and respective WT mice at the age of 14-16 weeks were used in FS experiments (fig. 5 c-f). C57Bl/6NCrI mice at the age of 13-15 weeks were used in FS/ ULK1 inhibitor experiments (fig. 5 g-j). Microdialysis experiments were performed during the first half of the day at an inverted day-night light cycle. FS was applied between 11.00, and 12.00 am.

(27) Explain the rationale for choosing one-tailed over two-tailed unpaired t tests.

Response:

We know that Dex treatment, i.e. GR activation, leads to FKBP51 induction and, therefore, we expect a change in only one direction.

(28) Author Contributions: Properly indicate which type of experiments were performed by KH, TB, JH, MLP and LJ. Please indicate contribution of Felix Hausch, Christoph W Turck, Alexandra Philipsen, Mathias V. Schmidt, and Bernhard Kuster.

Response:

Kathrin Hafner: performed Western blots of CoIP experiments, Thomas Bajaj: generated SIM-A9 KO cells, Felix Hausch: Synthesized SAFit1, Christoph W Turck: Analysed Interactome data, Alexandra Philipsen: statistical analyses, Jakob Hartmann, Mathias V Schmidt: genotyped/ breed FKBP51 KO Animals, Bernhard Kuster: Analysed secretomics data, Lee Jollans co-wrote and optimized the text-mining Python code.

(29) All references should be in the same format. (Some include web link and DOI, see #26 and 35)

Response:

The bibliography formatting has been revised and modified according to the feedback.

(30) Fig S3: Equal amount of Gapdh protein cannot be appreciated in the figure.

Response:

Quantifications of the blot were added and GAPDH quantifications are indicated in the following graph. Despite being the GAPDH signal slightly lower in the KO compared to the WT line, this difference is negligible compared to the difference in Atg5 signal in WT compared to KO, where in the KO line the signal is not detectable.

Reviewer #3

We thank Reviewer 3 very much for their positive and constructive feedback. Their interesting observations led us to further investigate and clarify the role of stress-primed secretory autophagy on the neurophysiological level.

Reviewer #3 (Remarks to the Author):

FKBP51 (gene name: FKBP5) is a glucocorticoid (GC) receptor binding protein, which acts as a co-chaperone of heat shock protein 90 (HSP90) and regulates GC-mediated stress. This protein is also known to be associated with mental disorders.

The authors of this study showed previously that GC-mediated stress leads to the activation of macroautophagy, which is regulated by FKBP51 (Gassen et al., *pLos Med.*, 2014). In the present study, they show that GC induces another type of autophagy called secretory autophagy and that FKBP51 plays an important role in this secretory autophagy by interacting with specific SNARE proteins (Fig. 1). They further show that MM9 is a novel cargo molecule of GC-mediated secretory autophagy (Fig. 3), that FKBP51 is critical for secretion of MMP9 (Fig. 4), and that the MMP secretion plays an important role in BDNF maturation (Fig. 4, 5).

The authors demonstrated these results using FKBP51 overexpressed HEK-293 cells, Atg5 KO and FKBP-5 KO cell lines as well as FKBP5 KO mice. The story is novel, logical and supported by an impressively large amount of data which appear to be sound and quite convincing. I have two concerns however.

- (1) The authors provide strong cellular evidence that FKBP51 is critical for secretory autophagy of MMP9 leading to BDNF maturation. However, I feel that the biological significance for this role of FKBP51 in living mice is missing or obscure.

As the authors stated in Abstract, BDNF is essential for synaptic plasticity. Thus, I would imagine that the reduced BDNF maturation in the stress response by FKBP5 KO has strong a strong behavioral phenotype in the BDNF-related behavior, such as learning and memory. However, previous study seems to fail to show such phenotype in the FKBP5 KO mice. For instance, O'Leary et al. (2011, pLos One) described that FKBP51 KO mice showed antidepressant behavior without affecting cognition and other basic motor functions. This previous result does not seem to be consistent with the proposed function of FKBP51 in stress-induced BDNF maturation. The authors of the present study should show some sort of behavioral or neurological phenotype associated with the reduced BDNF maturation by FKBP5 KO mice. Without such evidence, the readers of Nature Communications remain puzzled about the role of FKBP5 in stress-induced synaptic plasticity and BDNF maturation.

Response:

We agree with reviewer #3 that further investigation on the learning and memory effect would be of extreme interest. However, we think that acquiring such answers is out of the scope of this manuscript, but would rather represent the subject of an interesting follow-up project. The possible physiological and behavioral downstream effects could be many and variable. However, in order to shed some light on possible effects of increased mBDNF on neuroplasticity, we performed 2-photon experiments. The resulting data (Fig. 6) provide a validation to the physiological effect of stress-induced secretory autophagy not only on BDNF maturation but also on the consequential change of neuroplasticity in *ex vivo* murine organotypic brain slices. With the obtained results we could confirm our previous hypotheses and corroborate the effect on a neurological phenotype.

Regarding the absence of cognitive effects of the lack of FKBP51 reported by O'Leary et al., an important aspect to consider is that for that paper only old mice (between 17 and 22 months of age) were used. Age is a fundamental component when analyzing cognitive behaviors and the outcome of the same experiment might have been different in younger animals. The same is true for other variables such as type and intensity of stress. With our results we highlight the fact that there is a novel pathway that links excessive stress to neuroplasticity and we hypothesize that this is a mechanism regulating stress adaptation and possibly be correlated to psychiatric disorders when dysregulated. The exact consequences of such pathway need to be analyzed into detail and differentiated from other similar pathways triggered by similar but different stimuli.

- (2) Co-IP data in Fig. 1J is NOT described correctly. Fig. 1J is labeled as "GFP-IP", which I believe is correct. Based on the figure, I believed that the authors

overexpressed SEC22B as GFP-fusion protein and IPed using GFP-Ab. However, the used antibody was labeled as “FLAG-Ab”. This should be “GFP-Ab”. The Figure legend to Fig. 1J described this experiment as “FLAG-tagged FKBP51 co-IP (FLAG-IP). I believe that the legend should be “GFP-tagged SEC22B co-IP (GFP-IP).

Response:

Thank you for noticing. It is was indeed a labelling mistake and we corrected it.

Reviewer #4

We thank Reviewer #4 for their insightful feedback and suggestions. We addressed all the raised issues by performing additional experiments and by editing the manuscript in order to answer all the concerns.

Reviewer #4 (Remarks to the Author):

In this manuscript by Martinelli et al., termed “ Stress-primed secretory autophagy drives extracellular BDNF maturation” the authors identify the matrix metalloproteinase 9 (MMP9) as a stress-induced secreted protein involved in the cleavage of pro-brain derived neurotrophic factor (proBDNF) to its mature form (mBDNF). The authors demonstrate the involvement of the co-chaperone FK506-binding protein 51 (FKBP51) in stress-elevated secretion of MMP9 in the mouse brain, exploiting in vivo microdialysis in WT and Fkbp5 KO mice. The importance of the autophagy machinery for the stress elevated secretion is assessed in WT mice by including a ULK1 inhibitor. The authors claim that stress-induced secretion of MMP9 is through secretory autophagy, facilitated by FKBP51.

The novel finding in this manuscript is the involvement of FKBP51 in stress-elevated secretion of MMP9 in the mouse brain, resulting in maturation of BDNF. BDNF is essential for synaptic plasticity and altered BDNF signaling is associated with stress-related psychopathology. Hence, this finding is of general interest and contributes to the understanding of MMP9 secretion in the CNS. Overall, the biochemical data are well performed and the use of proteomic methods, data mining and in vivo microdialysis reflect an extensive amount of work. Nevertheless, the manuscript has some critical shortcomings that have to be addressed before publication.

The authors have previously reported involvement of FKBP51 in glucocorticoid-mediated activation of macroautophagy (Gassen et al., 2014; doi: 10.1371/journal.pmed.1001755). However, the presented data here for the involvement of secretory autophagy in stress- induced and FKBP51 dependent MMP9 secretion is not very convincing.

Firstly, the authors use cathepsin D (CTSD) and claim it is “ the established

secretory autophagy cargo” (page 5 line 117 and 118, page 12 line 287 and page 15 line 362). In the given reference for this claim (Kimura et al., 2017; doi: 10.15252/embj.201695081), CTSD is listed as a candidate secretory autophagy cargo in a proteomic screen. The established and most studied secretory autophagy cargo is interleukin-1beta (IL-1 β) (see e.g. New and Thomas 2019; doi: 10.1080/15548627.2019.1596479). It thus seems strange to not use IL-1 β and the authors should explain their choice. Interestingly, cathepsin B was shown to be secreted along with IL-1 β (Dupont et al., 2011; doi: 10.1038/emboj.2011.398). The authors do show the presence of CTSD in a co-IP of ectopically expressed FLAG-TRIM16 in SH-SY5Y cells (Fig. 1e). However, that does not necessarily mean that TRIM16 binds to CTSD since IP-based interaction detection can reflect indirect binding of the proteins in question.

Secondly, FKBP51 is a known co-chaperone of heat shock protein 90 (HSP90) and HSP90 has been assigned a key role in secretory autophagy of IL-1 β , mediating import of IL-1 β into the autophagosomal intermembrane space (Zhang et al., 2015; doi: 10.7554/eLife.11205). The authors do not address this point at all other than showing that mutating the HSP90 binding site of FKBP51 reduces a potential interaction between FKBP51 and galectin 8 (Fig. 2b). Finally, the authors show that FKBP51 is essential for the association of SEC22B with its Q-SNARE partners in SH-SY5Y cells (Fig. 1j). However, there are no data presented that demonstrate the importance of the formation of this SNARE complex for stress-induced MMP9 secretion in SIM-A9 (microglia) cells or mice. Detailed comments and suggestions are included below.

Response:

Concerning the use of CTSD instead of IL-1b as an established cargo, this was done for two reasons:

1) the implication of IL-1b in this pathway is the focus of another study we are currently completing (Hartmann *et al.*, in preparation). We attach here confidential results showing that IL-1b is regulated in the same way as CTSD.

Quantification of IL1b via ELISA assay. IL1b from supernatants was measured via ELISA after SIM-A9 cells were treated as follow a) LLOMe for 4, 8 and 24 hours or vehicle for 24 hours. b) 3nM, 30nM and 300nM Dex or vehicle for 4 hours. c) 300nM Dex or vehicle for 4 hours in WT and Atg5 KO SIM-A9 cells. d) transfected with FKBP51 expressing plasmid or control vector. *P < 0.05; ***P < 0.001; ****P < 0.0001. Tukey's multiple comparison test was used for a, b and c; unpaired t-test was used for d. Significances in c are referred to comparison of Dex 300nM with each of the other conditions. Error bars expressed in SEM.

2) IL1b was not detected as part of the secretome in the MS experiment, probably because below the detection limit. Therefore, to give a complete picture, we decided to opt for CTSD as another well-characterized secretory autophagy cargo.

Finally, regarding the importance of the SNARE complex formation for the stress-induced MMP9 secretion, we generated a SIM-A9 SEC22B KO line with which we demonstrated that the absence of SEC22B impairs not only the secretion of MMP9, but also of CTSD. Detailed results are shown in response to comment 6.

Major comments:

- (1) Figure 1b,c,d and e and figure S1 a and b: Interaction between FKBP51 and SEC22B or TRIM16 is implicated from reciprocal co-IPs. To demonstrate a direct binding between FKBP51 and SEC22B or TRIM16 the authors could use a GST-pulldown assay with labeled *in vitro* translated proteins. Furthermore, co-localization images in cells to visualize FKBP51 together with SEC22B or TRIM16 would be helpful. The same applies for the implied interaction between TRIM16 and CTSD.

Response:

To address this point (addressed also by Reviewer #2), we performed the suggested pull-down experiments to verify the direct interaction between FKBP51 and TRIM16 and found that there is indeed a direct interaction *in vitro* (supplementary Fig. S1c). However, we also rephrased the parts describing these results replacing the term “interaction” with “association”, meaning an either direct or indirect interaction. In fact, for the presented mechanism it is irrelevant whether the interaction is direct or not. What we show is that the proteins form a complex for which they are immunoprecipitated together, and, more importantly, this association (whether it is direct or not) affects the pathway.

- (2) For the blots in figure 1d, e and g, which form of CTSD is shown/recognized by the antibody? CTSD exists in different forms with the inactive precursor of the enzyme, procathepsin D, being cleaved, resulting in different forms of mature/active cathepsin D.

Response:

As shown in the representative blot below, the predominant and quantified CTSD form is the cleaved/mature one (CTSD heavy chain).

- (3) Figure 1I- “Schematic overview of the interactions of FKBP51 in the secretory autophagy pathway”: Here FKBP51 is shown to interact with GAL8 but no data have yet been presented to show this. TRIM16 binds to GAL8 and the figure should indicate that. Furthermore, HSP90 is shown as a binding partner of FKBP51 but HSP90 is not included in the blots of any of the IPs in figure 1. Furthermore, there are no data

presented indicating the importance of FKBP51 for transfer of the TRIM16-cargo (CTSD) to the autophagosome or data indicating that HSP90 is not present. TRIM16 association to SEC22B is independent of FKBP51 according to figure 1e. And there are no comments or experiments addressing how CTSD, that normally resides in lysosomes, is translocated into the lumen of autophagosomes prior to its secretion. Therefore, the claim on page 6 line 148-149. "From these data, FKBP51 results to be involved in several key steps of the secretory autophagy pathway (Fig 1l)", appears as an overstatement.

Response:

With the FKBP51-IP displayed in Fig 2b, we show that FKBP51 associates with GAL8. Whether this association is direct or indirect is irrelevant for the proposed mechanism. In fact, we state that this association is, at least partially, indirect and occurs via HSP90. With additional WB analyses we detected HSP90 in the FKBP51 eluate, as further confirmation of our hypothesis (these data have been added to Fig. 2b). The model represented in Fig 1l, takes into account data from Fig. 2 (as stated in the image). Taken together these data we, therefore, do not think that asserting that FKBP51 is involved in several key steps of secretory autophagy is an overstatement because being involved represents a very mild action verb and, for that meaning, we believe to have the adequate supporting data.

- (4) In figure 2c and 2d the authors use a tandem tagged (mRFP-GFP) galectin 3 (tfGal3) in SH-SY5Y cells to monitor lysosomal damage. The reduction of the GFP signal is a result of acidification of tfGal3. Gal3 is recruited to damaged lysosomes and Gal3 becomes acidified through lysophagy. Lysophagy involves autophagosomal engulfment of damaged lysosomes that subsequently become degraded by fusion with intact lysosomes (Maejima et al., 2013; doi: 10.1038/emboj.2013.171). LLOMe induces lysosomal damage that culminates in lysophagy and Dex appears to be able to do the same. Inhibition of lysosomal acidification with BafA1 abolishes the effect. Therefore these data actually show degradation of Gal3 on damaged lysosomes through autophagy. The authors should comment on how they envision the effect of dex on lysosomes and how this relates to secretory autophagy.

Response:

With our study, we do not expect to answer the question of how Dex can lead to lysosomal damage (e.g. lysosomal membrane permeabilization) as it would be a far too complex topic to investigate and is beyond the interest of this manuscript. However, with our data we can confidently affirm that Dex indeed leads to lysosomal damage and activates the repair mechanism involving the recruitment of galectines on the lysosomal membrane. As for the mechanism leading to the lysosomal damage, many could be the hypotheses, but having no data to this regard we are reluctant to postulate any. Here are some articles that describe the mechanisms (and its complexity) that can lead to lysosomal damage and that are still not fully unraveled: Jia

et al., Galectins Control mTOR in Response to Endomembrane Damage, Mol. Cell, 2018; Napolitano and Ballabio, TFEB at a glance, J. cell sci., 2016. A hypothesis of a more direct effect of GCs on the biophysical properties of cell membranes, that we can apply to the mechanism leading to lysosomal damage, was described by Van Laethem et al. (J Immunol, 2003). In their study, dexamethasone caused alterations in lipid raft palmitate content inducing a decline in the proportion of saturated fatty acids while increasing unsaturated ones. From a biophysical perspective, the changes in membrane lipid composition increase fluidity and therefore it is tempting to speculate that these changes also affect lysosomal osmotic stability (Yang et al., Cell Biol. Int., 2013).

- (5) In figure 2 e) and 2 f) SIM-A9 secretion of CTSD in response to LLOMe and dex treatment, respectively should also include BafA1 treatment to determine if the secretion is dependent on functional lysosomes or not.

Response:

We performed additional experiment that were added to supplementary Fig. S2 as panel c, where we show that Baf has the expected effect on CTSD secretion (i.e. reversion of Dex and LLOMe effects) and confirmed that CTSD secretion correlates indeed with lysosomal damage.

- (6) In figure 4 the dex induced MMP9 secretion should be shown in FKBP51 KO SIM-A9 cells as well to complement the in vivo results in figure 5. Furthermore, in order to link MMP9 secretion to secretory autophagy the authors could use siRNA knockdown of TRIM16 or SEC22B in these cells. The presence of MMP9 in an IP of TRIM16 or co-localization study of MMP9 with TRIM16 in cells would also be desirable.

Response:

To answer this question, we generated *Fkbp5*-KO and *Sec22b*-KO SIM-A9 cells and analyzed CTSD, MMP9, proBDNF and mBDNF secretion via WB of supernatants. In line with the rest of our data, results of this experiment showed that the secretion of CTSD, MMP9 and mBDNF is both SEC22B and FKBP51 dependent as is significantly impaired in the both KO cell lines compared to WTs, while the secretion of proBDNF is unaffected by FKBP51 or SEC22B (see Fig. S4 f and g).

- (7) Figure 7-“Schematic representation of the findings and proposed model”: In a, TRIM16 is shown to interact with MMP9. Again, there are no data in the manuscript that demonstrate this interaction and this schematic drawing is thus not accurate.

Response:

We rephrased the figure title as “proposed model based on the findings”.

Minor comments:

- (8) In Supplementary Table S1 of 29 identified interactors of FKBP51 in HEK293 cells, HSP90 is not listed. The authors should comment on that.

Response:

The detection method used for the Interactome analysis was very stringent. Only proteins found to interact with FKBP51 and not with the control plasmid were considered interactors. HSP90 was found both in the FKBP51 and control IPs and therefore excluded from the final list, probably as a false negative since it is a well known FKBP51 interactor.

- (9) Page 5 line 114-115: "...the interaction of FKBP51 with SEC22B (Fig. 1b,c), previously only deduced via differential centrifugation¹³". Please clarify, since the cited reference (Kimura et al., 2017; doi: 10.15252/embj.201695081) appears not to have any data implying interaction between FKBP51 and SEC22B.

Response:

We apologize for the mistake. It was meant to be referred to the association between TRIM16 and SEC22B. We corrected the mistake in the text.

- (10) In figure 1g, FKBP51 levels are not increased with 100 nm Dex. This is in contrast to supplementary figure 1d showing quantified increase of FKBP51 expression with 100 nm Dex. The authors should comment on this and include an image of a representative blot for the figure S1d.

Response:

We replaced the blot with a more representative one (also used originally for the quantification), where the FKBP51 increase can be appreciated by eye.

- (11) In figure 2a the authors should include a representative image of the blot.

Response:

We added a representative image of the western blot to figure 2a, furthermore all uncropped blots will be made available in the source data file.

- (12) The figure legends should indicate which cell type is used. This is lacking in several of the figure legends and should be clearly indicated since both HEK293, SH-SY5Y and SIM-A9 cells are used.

Response:

All missing information was added.

- (13) Methods-ELISA page 22. In the description, ELISA kit product numbers are indicated and not antibody product numbers as the text indicates. This should be corrected. Furthermore, in line 557 it says "Amounts of Il1b were detected with a plate reader". Il1b is not included in this study.

Response:

We apologize for the mistake regarding Il1b, we corrected it and provided the right information.

Regarding the ELISA kits, the products already contain the antibodies. We rephrased the description for clarity.

- (14) In the figure legend to figure 2, page 40 line 028, dexamethasone should be written after 300 nM.

Response:

Thank you for noticing, we added the information.

- (15) In figure legend to figure 5 page 41, there is a typo error in line 056. ILK1 inhibitor, should be ULK1 inhibitor.

Response:

Thank you for noticing, the mistake was corrected.

- (16) The discussion section is rather long and should be more focused on the obtained data in this study.

Response:

We thoroughly revised the discussion and adapted it to the new data with this comment in mind.

Reviewer #5

We thank reviewer 5 very much for their supportive feedback. We addressed their insightful concerns with the hope that we could clarify our proteomics findings.

Reviewer #5 (Remarks to the Author):

The manuscript by Martinelli et al. "Stress-primed secretory autophagy drives extracellular BDNF maturation" described the mechanism of enhancement of secretory autophagy by glucocorticoid-induced stress. The authors use interactomics and secretome analysis by mass spectrometry to identify proteins involved in the process and propose an elegant step-by-step mechanistic model validated by several other methods. The paper is well written, the findings are novel and this reviewer supports the publication in Nature Communications which will allow these results to reach a broad readership. There are a few minor concerns, mostly technical in nature, that should be addressed before acceptance.

Detailed comments:

- (1) Since TRIM16 was confirmed to be an interactor of FKBP51 by Western blot, but not originally found in the MS dataset, can the MS data be researched and perhaps TRIM16 peptides can be found (maybe with the help of the inclusion list?)

Response:

TRIM16 was not found in the FKBP51 interactome list, nor in the control one. We think that this is due to a below the detection level expression of this protein. In fact, we confirmed the interaction of FKBP51 with TRIM16 via a pull down assay (Supplementary Fig. S1c).

- (2) For the interactome analysis, the transfection with a vector containing FLAG only was used. Can the authors include more description of how the control (unspecific) binders were eliminated? The only explanation I could find is in lines 651-651, that the proteins overlapping from all four replicates were counted as interactors. This is not sufficient.

Response:

We considered as interactors of FKBP51 only proteins that fulfilled **both** of the following criteria:

- Bind only to FKBP51-FLAG and not control FLAG
- Be found in all four replicates of FKBP51-FLAG transfected cells

This selection method is quite stringent as it results in the exclusion of false negatives, but allows us to be confident in the identification of the positive candidates.

- (3) For the secretome analysis, the media supplemented with FBS was used throughout the whole experiment. Is that correct? If so, how was the signal suppression by the overwhelming amount of protein handled? Was the albumin removed? This was not an issue for detection because the labeling was used, but signal suppression would be an issue anyway. Can the authors comment on that?

Response:

Yes, the secretomics experiment has been carried out in the presence of FBS to avoid any artefacts introduced by serum starvation. In order to study secreted proteins of low intensity irrespective of e.g. high-intense albumin in the background, we performed labeling of newly synthesized proteins and selective enrichment according to a protocol that has been published before (Eichelbaum K, Winter M, Diaz MB, Herzig S, Krijgsveld J: Selective enrichment of newly synthesized proteins for quantitative secretome analysis. *Nat Biotech* 2012, 30(10):984-990.). In more detail, the cells were labeled with azide-containing azidohomoalanine that is substituting methionine in newly synthesized proteins. In a second step, these proteins were selectively enriched and covalently linked to alkyne beads by click chemistry. Consequently, other non-labeled proteins such as FBS-derived albumin were removed before subsequent LC-MS/MS analysis was performed.

- (4) Another useful clarification of the secretome analysis would be the comparison of growth rates and cell death between the wild type and the Atg5 KO cells. Are they identical in this respect, and if not, how was the data normalized?

Response:

For the accurate quantification of secreted proteins compared between WT vs. Atg5 KO cells, we applied a method called 'intensity-based absolute quantification' (iBAQ; Schwanhaussner, B. et al. Global quantification of mammalian gene expression control. *Nature* 473, 337-342, doi:10.1038/nature10098 (2011)) in the MaxQuant software that serves as an estimate for absolute quantification. By using this way of protein quantification, summed intensities of peptides matching to a specific protein are divided by the number of theoretically observable peptides, providing an accurate proxy for protein intensity. A further normalization between both samples - WT vs. Atg5 KO - has not been carried out since a comprehensive perturbation of the secretory pathway has been expected that other normalization tools (e.g. total sum normalization) used commonly in proteomics were rejected.

REVIEWERS' COMMENTS

Reviewer #1 (Remarks to the Author):

The revised version as well as the detailed reply answers all my concerns. This is an excellent contribution to the field of BDNF processing in the central nervous system. The study opens new gates in the field of plasticity-related secretory proteins in the CNS. I'd like to recommend publication in Nature Communications.

Reviewer #2 (Remarks to the Author):

This revised article describes a novel and broadly interesting role for FKBP51 in regulating stress-related secretory autophagy. The authors use multiple complementary models and sound methods to support their claims. The authors should be commended for their thorough attention to all of the prior concerns. The revised manuscript has greatly improved the clarity of the figures and the text and now convincingly support the conclusions. This important work is the first description of MMP9 as a cargo of secretory autophagy and that this triggers maturation of BDNF. There are no additional concerns to be addressed.

Reviewer #3 (Remarks to the Author):

To address my major concern about the biological significance of secretory autophagy by FKBP51, the authors performed new 2-photon experiments. The resulting data (Fig. 6) supported the physiological effect of stress-induced secretory autophagy not only on BDNF maturation but also on the consequential change of neuroplasticity (increased spine density) in ex vivo murine organotypic brain slices. The authors also corrected the mislabeling of Fig. 1J, which was my minor concern. Therefore, I agree that the authors fully addressed my concerns.

Reviewer #4 (Remarks to the Author):

Overall the authors have adequately addressed the comments and concerns of the referees. The authors have done a considerable effort by including a number of new experiments and analyses in the revised version resulting in a greatly improved manuscript. The authors demonstrate an effect of MMP9 mediated BDNF maturation on synaptic plasticity using organotypic hippocampal slice cultures. Furthermore, the authors show that secretion of CTSD, MMP9 and mBDNF is dependent on FKBP51 as well as on SEC22B using KO SIM-A9 cells. However, the link to secretory autophagy would have benefited from co-localization/cell imaging studies and demonstration of direct interactions between TRIM16 and the putative secretory autophagy cargos CTSD and MMP9, respectively. In addition, the authors actually show that secretion of CTSD is dependent upon functional lysosomes/lysosomes with a low pH (BafA1 treatment, figure S2). Thus, the lysosome is not bypassed. In their rebuttal response to my specific point (5) regarding this new figure, the authors claim: "Baf has the expected effect on CTSD secretion (i.e. reversion of Dex and LLOMe effects) and confirmed that CTSD secretion correlates indeed with lysosomal damage." This assumption is wrong because BafA1 affects the pH level of lysosomes and cannot revert the effects of LLOMe that results in rupture of lysosomes. BafA1 on the other hand inhibits re-acidification (restoration of a low pH) in repaired lysosomes and inhibits degradation of engulfed autophagic cargo such as severely damaged lysosomes (that are damaged beyond repair and destined for lysophagy). Therefore, an alternative pathway for CTSD secretion that involves functional lysosomes cannot be excluded. A paper by Padamsey et al. 2016 (<https://doi.org/10.1016/j.neuron.2016.11.013>) shows that lysosomes act in synaptic plasticity by undergoing regulated fusion with the plasma membrane to release enzymes (including cathepsin B involved in activating MMP9) that enable dendritic remodeling.

The authors should modify the result section (lines 221-222) regarding BafA1 inhibition of CTSD secretion accordingly and address this in the discussion section and figure 7a. Minor concerns are listed below.

Specific comments:

- (1) In legends to figure 1, figure 2 and figure S2 a-c the cell type used should be specified.
- (2) In figure 1g it seems strange that there is no elevation in the amount of FKBP51 with Dex 100 nM treatment (WCE blot) as demonstrated for FKBP51 in the blot in j.
- (3) In the legend to figure 1i the authors should clearly indicate that the figure is an illustration/model of how the authors envision CTSD secretion.
- (4) CTSD is not an established secretory autophagy cargo and even though the authors have unpublished data on the same mechanism for IL-1 β they should in this manuscript refer to CTSD as a putative secretory autophagy cargo when first mentioned in line 134-135.
- (5) In the legend to figure 2, line 1227 the concentration of LLOMe used is lacking.
- (6) In the legend to figure S2 there is a typo error in lines 21 and 27 "SIMA-9 cells" instead of SIM-A9.
- (7) In the legend to figure S2, line 26 the time used for 300 nM Dex or 50 nM LLOMe treatment should be indicated.
- (8) The authors sometimes write Fkbp51 KO instead of Fkbp5 when referring to the gene both in the text (e.g. line 311) and figures and this should be corrected.
- (9) In figure S4d and e the blots for KO of FKBP51 and SEC22b look more like knockdowns than knockouts since a detectable amount of the proteins is clearly visible. The authors should comment on this. Was sequencing of KO clones conducted?
- (10) In figure 6b-e and figure S6a-d the y-axis title is lacking.
- (11) The text in lines 390-400 describing figures 6f-h is a bit unclear, at least to a reader that is unfamiliar with the assay. The method section on this is also very limited. How is a "change in spine density" determined and how do you measure "increased spine density". Is that term the same as "an increase in spine formation"? Are the differences between the groups in h significant?
- (12) Not clear how figure 6Sf correlates with figures 6g and h
- (13) In the discussion, the authors write (line 422-423): "...damage of lysosomal membranes, which in turn causes a switch from macroautophagy to secretory autophagy". The results in this manuscript do not provide any evidence for such a switch.
- (14) Discussion line 425-426: "FKBP51 is recruited to damaged lysosomes where it bridges the fusion to autophagosomes via SEC22B". Fusion of what?

Reviewer #5 (Remarks to the Author):

The authors have addressed my comments in an adequate manner.

Rebuttle Reviewer #4

Reviewer #4 (Remarks to the Author):

Overall the authors have adequately addressed the comments and concerns of the referees. The authors have done a considerable effort by including a number of new experiments and analyses in the revised version resulting in a greatly improved manuscript. The authors demonstrate an effect of MMP9 mediated BDNF maturation on synaptic plasticity using organotypic hippocampal slice cultures. Furthermore, the authors show that secretion of CTSD, MMP9 and mBDNF is dependent on FKBP51 as well as on SEC22B using KO SIM-A9 cells

However, the link to secretory autophagy would have benefited from co-localization/cell imaging studies and demonstration of direct interactions between TRIM16 and the putative secretory autophagy cargos CTSD and MMP9, respectively. In addition, the authors actually show that secretion of CTSD is dependent upon functional lysosomes/lysosomes with a low pH (BafA1 treatment, figure S2). Thus, the lysosome is not bypassed. In their rebuttal response to my specific point (5) regarding this new figure, the authors claim: "Baf has the expected effect on CTSD secretion (i.e. reversion of Dex and LLOMe effects) and confirmed that CTSD secretion correlates indeed with lysosomal damage." This assumption is wrong because BafA1 affects the pH level of lysosomes and cannot revert the effects of LLOMe that results in rupture of lysosomes. BafA1 on the other hand inhibits re-acidification (restoration of a low pH) in repaired lysosomes and inhibits degradation of engulfed autophagic cargo

such as severely damaged lysosomes (that are damaged beyond repair and destined for lysophagy). Therefore, an alternative pathway for CTSD secretion that involves functional lysosomes cannot be excluded. A paper by Padamsey et al. 2016 (<https://doi.org/10.1016/j.neuron.2016.11.013>) shows that lysosomes act in synaptic plasticity by undergoing regulated fusion with the plasma membrane to release enzymes (including cathepsin B involved in activating MMP9) that enable dendritic remodeling. The authors should modify the result section (lines 221-222) regarding BafA1 inhibition of CTSD secretion accordingly and address this in the discussion section and figure 7a. Minor concerns are listed below.

Specific comments:

- (1) In legends to figure 1, figure 2 and figure S2 a-c the cell type used should be specified.
- (2) In figure 1g it seems strange that there is no elevation in the amount of FKBP51 with Dex 100 nM treatment (WCE blot) as demonstrated for FKBP51 in the blot in j.
- (3) In the legend to figure 1i the authors should clearly indicate that the figure is an illustration/model of how the authors envision CTSD secretion.
- (4) CTSD is not an established secretory autophagy cargo and even though the authors have unpublished data on the same mechanism for IL-1 β they should in this manuscript refer to CTSD as a putative secretory autophagy cargo when first mentioned in line 134-135.
- (5) In the legend to figure 2, line 1227 the concentration of LLOMe used is lacking.
- (6) In the legend to figure S2 there is a typo error in lines 21 and 27 "SIMA-9 cells" instead of SIM-A9.
- (7) In the legend to figure S2, line 26 the time used for 300 nM Dex or 50 nM LLOMe treatment should be indicated.
- (8) The authors sometimes write Fkbp51 KO instead of Fkbp5 when referring to the gene both in the text (e.g. line 311) and figures and this should be corrected.
- (9) In figure S4d and e the blots for KO of FKBP51 and SEC22b look more like knockdowns than knockouts since a detectable amount of the proteins is clearly visible. The authors should comment on this. Was sequencing of KO clones conducted?
- (10) In figure 6b-e and figure S6a-d the y-axis title is lacking.
- (11) The text in lines 390-400 describing figures 6f-h is a bit unclear, at least to a reader that is unfamiliar with the assay. The method section on this is also very limited. How is a "change in spine density" determined and how do you measure "increased spine density". Is that term the same as "an increase in spine formation"? Are the differences between the groups in h significant?
- (12) Not clear how figure 6Sf correlates with figures 6g and h
- (13) In the discussion, the authors write (line 422-423): "...damage of lysosomal membranes, which in turn causes a switch from macroautophagy to secretory autophagy". The results in this manuscript do not provide any evidence for such a switch.
- (14) Discussion line 425-426: "FKBP51 is recruited to damaged lysosomes where it bridges the fusion to autophagosomes via SEC22B". Fusion of what?

Answers:

We thank Reviewer 4 for their feedback and constructive remarks.

We agree that a co-localization study could have been very interesting. However, we do not think that the lack of co-localization evidence lessens the functional findings, as also stated in our previous rebuttal letter.

Regarding a possible alternative pathway, we completely agree with reviewer 4 that our proposed model is not the only secretory pathway involved in neuronal plasticity, as clearly expressed throughout the manuscript. We rather propose this one as a novel mechanism linked to lysosomal damage. With the Gal3 experiments (figure 2) we see that increased Dexamethasone concentrations lead to increased lysosomal damage. At the same time, increased Dexamethasone concentrations lead to increased CTSD secretion (ELISA assays). Together with the Immunoprecipitation assays, this strongly suggest that it is a mechanism depending on damaged lysosomes rather than intact ones. In fact, with our experiments we demonstrated that Baf1A is indeed able to prevent the re-acidification of lysosomes and reduces CTSD secretion to vehicle-levels (Supplementary Fig. S2c). Furthermore, the Padamsey et al. publication, refers to activity dependent secretion in neurons, which is not a system we used. We observed an activity independent release in cell lines. We believe that there are different mechanisms that contribute to the synaptic plasticity and that involve the secretion of key molecules. With this study we showed that secretory autophagy is one of the mechanisms that leads to synaptic plasticity triggered by glucocorticoid-mediated stress. We also find it highly probable that different stimuli induce synaptic plasticity through different pathways (such as neuronal activity via lysosomal fusion to the membrane vs. stress via secretory autophagy for example). Further studies on the interaction of stress-mediated secretory autophagy with neuronal activity, would for sure be very interesting and could constitute good follow-up projects.

- (1) We now provide information about celltypes used in experiments, in the corresponding figure legends.
- (2) We exchanged the blot in Figure 1g with a representative blot showing a more pronounced band profile for FKBP51.
- (3) We now clearly indicated in the figure legend for figure 1i that we present a schematic model depicting the proposed molecular mechanism.
- (4) We now introduce CTSD as a putative secretory autophagy cargo protein in the main text.
- (5) We now indicate the concentration of LLOMe used in figure legend of figure 2.
- (6) We corrected the typo.
- (7) We now indicate the duration time of Dexamethasone or LLOMe incubation in figure legend of supplemental figure 2.
- (8) We corrected all mis-spelled gene names and protein names and made the spelling for Fkbp5 vs Fkbp51 consistent throughout the manuscript.
- (9) We now provide more convincing blots from *Sec22b* and *Fkbp5* KO cells using PA gels with better separation and a new antibody for detecting FKBP51 showing less background interference. The new blots clearly show that the cells used in were knockout.
- (10) We corrected the labelling of y-axes for figure 6 b-e and supplemental figure 6 a-d.
- (11) In the method section (p31, lines 754ff) we describe the count of spines per length of dendritic segment (3-5/slice) over time. Although we believe that we included all necessary information about the data acquisition in the method section, indeed a re-phrasing could facilitate understanding as we also noted that the observation unit "dendrite" as used in the results section was not pointed out well in the method part. We now write:
"Two-photon image analysis: A custom MATLAB® script was used to count and track the fate of individual spines on 3-5 individual dendritic segments (length >30 µm) in each imaged OHSC at 0 and 30 min (**t0, t30**). Spine density was calculated for each slice as the total number of spines per total length of the analyzed dendritic segments **for each time point**. Spine density at t30 was then and normalized to baseline (0 min) **at t0, indicating the spine density change then used** for statistical analysis.
- Statistics:* Three different OHSC preparations were used and conditions randomly distributed between cultures from individual pups. Spine density change **calculated as described in the previous section** was transformed to categorial classes (increased/decreased/unchanged spine density) to enable analysis of responsiveness of spine dynamics to the different treatments by Chi-squared statistics."
- (12) The reviewer also asked whether "an increase in spine formation" is identical for the term "increase in spine density". Indeed, the increase of spine density at t3 compared to t0 can only be explained by formation of new spines.
- (13) This sentence is indeed hypothetical and misleading. We changed it to: "...damage of lysosomal membranes, which in turn lead to an increase of secretory autophagy".
- (14) For more clarity, the sentence was deleted.